# Fix False Transparency by Noise Guided Splatting

**Aly El Hakie**[1]*,  **Yiren Lu**[2]*,  **Yu Yin**[2],  **Michael Jenkins**[1,2],   **Yehe Liu**[1,2]

[1]OpsiClear LLC
[2]Case Western Reserve University
[1]{aly, yehe}@opsiclear.com
[2]{yiren.lu, yu.yin, mwj5}@case.edu
https://opsiclear.github.io/ngs/

## Abstract

Opaque objects reconstructed by 3D Gaussian Splatting (3DGS) often exhibit a falsely transparent surface, leading to inconsistent background and internal patterns under camera motion in interactive viewing. This issue stems from the ill-posed optimization in 3DGS. During training, background and foreground Gaussians are blended via $\alpha$-compositing and optimized solely against the input RGB images using a photometric loss. As this process lacks an explicit constraint on surface opacity, the optimization may incorrectly assign transparency to opaque regions, resulting in view-inconsistent and falsely transparent output. This issue is difficult to detect in standard evaluation settings (*i.e.,* rendering static images), but becomes particularly evident in object-centric reconstructions under interactive viewing. Although other causes of view-inconsistency, such as popping artifacts, have been explored previously, false transparency has not been explicitly identified. To the best of our knowledge, we are the first to quantify, characterize, and develop solutions for this "false transparency" artifact, an under-reported artifact in 3DGS. Our strategy, Noise Guided Splatting (NGS), encourages surface Gaussians to adopt higher opacity by injecting opaque noise Gaussians in the object volume during training, requiring only minimal modifications to the existing splatting process. To quantitatively evaluate false transparency in static renderings, we propose a novel transmittance-based metric that measures the severity of this artifact. In addition, we introduce a customized, high-quality object-centric scan dataset exhibiting pronounced transparency issues, and we augment popular existing datasets (*e.g.,* DTU) with complementary infill noise specifically designed to assess the robustness of 3D reconstruction methods to false transparency. Experiments across multiple datasets show that NGS substantially reduces false transparency while maintaining competitive performance on standard rendering metrics (*e.g.,* PSNR), demonstrating its overall effectiveness.

## 1   Introduction

3D Gaussian Splatting (3DGS) [1] is an emerging neural rendering technique offering unprecedented real-time performance and fidelity through explicit scene representation. However, due to its unconstrained optimization and the $\alpha$-blending process, opaque surface Gaussians can be incorrectly learned as transparent. We define this largely unacknowledged artifact as false transparency.

False transparency causes opaque surfaces to incorrectly appear semi-transparent, especially observable during interactive viewing and undetectable in individual frames and with standard Image Quality Assessment (IQA) metrics. During camera movement, objects exhibit a disturbing "see-through"

---

*    These authors contribute equally.

39th Conference on Neural Information Processing Systems (NeurIPS 2025).

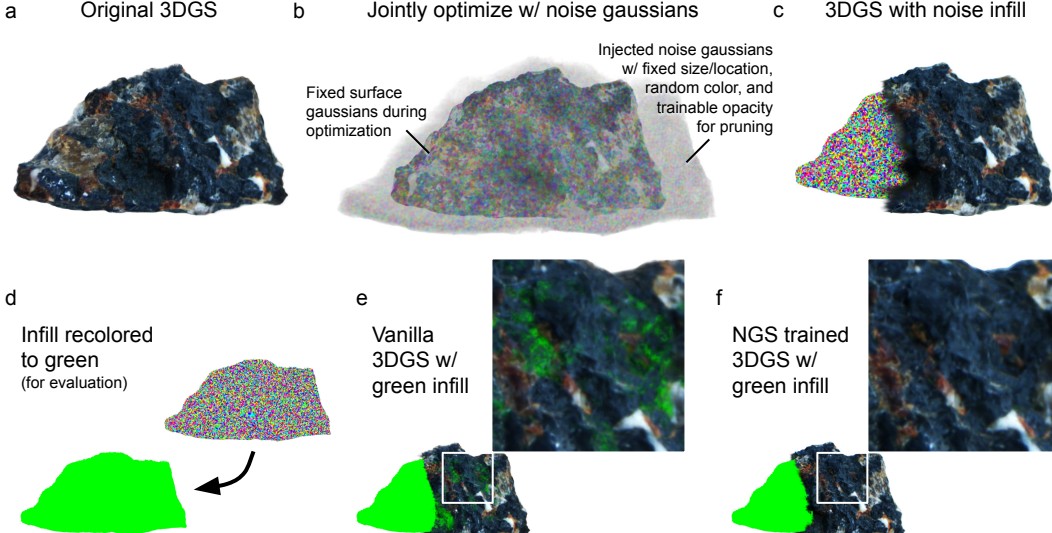

Figure 1: Overview of the NGS. (a) Object-centric 3DGS render of a stone. (b) Noise Gaussians are introduced to the training process. (c) Only visible noise Gaussians are removed during optimization, leaving subsurface noise Gaussians filling the object. (d) Noise infill can be recolored and saved for transparency evaluation. (e) Recolored infill inserted to the vanilla 3DGS revealing highly transparent regions on the surface. (f) Recolored infill does not leak through the NGS trained surface.

quality where internal and background Gaussian structures become visible through surfaces that should be opaque. These internal structures move out of alignment with the surface under changes in camera pose, creating an illusion reminiscent of frosted glass.

This false transparency artifact occurs because 3DGS is supervised primarily by a 2D photometric loss between rendered and ground-truth images. Such supervision creates ambiguity regarding true surface opacity, as the optimization can jointly refine foreground and inner background elements in discrete views, with their combined rendering still satisfying the 2D constraints. It is most commonly observed in regions lacking adequate visual cues to differentiate between an opaque surface and a semi-transparent surface backed by another surface, specifically in areas presenting low texture, repeating patterns, specular highlights, or geometric complexity.

The problem is also more pronounced in object-centric reconstruction. In scene-level reconstruction, many objects are not imaged from all angles and lack information about their posterior surfaces, resulting in less ambiguity. However, in 360° object-centric settings, the mean depth between opposing surfaces is small. At every angle, front-facing surface of the object is paired and jointly optimized with back surface along the same ray. Beyond visual artifacts, false transparency affects downstream applications such as surface extraction, physics simulations, and volumetric analysis. Many of such applications rely on accurate opacity to delineate object boundaries. An ambiguous alpha causes invalid or unreliable results.

In practice, evaluating the severity of this false transparency is challenging during both training and post-training analysis, as conventional metrics generally rely on comparing static renderings with a 2D reference. Some recent studies [2, 3, 4, 5, 6] have indirectly mitigated this issue. Although these studies advanced approximation techniques in the splatting process, for example by refining depth-ordering and $\alpha$-blending, their main goal was to improve view-consistency in 3DGS. As such, they did not directly address the underlying mechanisms responsible for false transparency.

This paper introduces Noise Guided Splatting (NGS) (Fig. 1), a novel strategy to address the opacity ambiguity during optimization by injecting persistent internal noise Gaussian structures within an object's volume, effectively enforcing surface opacity. NGS allocates high-opacity noise Gaussians with continuously randomized coloration within the object's volume. The infill creates an effective occlusion barrier between opposing surfaces, preventing the optimization process from integrating back surfaces into the front-facing rendering. Additionally, the noise points can be extracted and

used as a diagnostic tool to support evaluating false transparency when assessing any splatting-based rendering methods.

In summary, we make the following contributions:

1. A new technique that places interior noise Gaussians to distinguish between the interior and exterior space of the rendered object, guiding optimization toward proper surface opacity. The method is plug-and-play and requires minimal modifications to existing frameworks.
2. A new approach to visualize and quantify the false transparency in static 3DGS renderings.
3. A noise Gaussian infill dataset, including infill add-ons to existing datasets (e.g., DTU) and a customized high-resolution object-centric scan dataset, to facilitate benchmarking.

## 2   Related works

**Novel View Synthesis (NVS).** Neural Radiance Fields (NeRF) [7] revolutionized NVS through neural implicit representations for 3D scenes, achieving high visual quality. However, NeRF's computational demands led to the development of faster alternatives like sparse voxel grids [8] and hash encoding [9]. 3DGS [1] marked a major advancement through explicit trainable primitives, enabling real-time rendering with high fidelity. Building upon 3DGS, numerous works have enhanced its capabilities in numerous directions, including more efficient training strategies [10, 11], better densification heuristics [12], anti-aliasing [13] and reduced dependency on initialization [14].

**NVS artifacts.** There are several well recognized NVS artifacts that should not be confused with false transparency artifacts. A common one is floater artifacts, which manifest as sparse features reconstructed at incorrect depths above the surface [15, 16]. These artifacts do not appear in training views but become obvious in novel views. Floaters can be mitigated using depth consistency constraints [15, 17] and specialized priors [16, 18, 19]. Another category is view-inconsistency artifacts, which cause surfaces to exhibit unnatural changes during viewpoint transitions. A well-known example is the 'popping artifact' [6], caused by sorting discontinuities between adjacent views. Hierarchical sorting [6], order-independent transparency [2], anti-aliasing filtering [4], and hybrid transparency [5] have been introduced to address these problems. Finally, 3DGS also suffers from poor reconstruction of certain details, which has been addressed using specialized loss functions [20] and diffusion-based post-processing enhancements [21].

**Transparency.** Blending semi-transparent primitives is an essential feature of 3DGS, ensuring rendering fidelity and smooth transitions across viewing angles. Most research on 3DGS transparency focuses on accurately reconstructing inherently transparent objects [22, 23, 24]. However, the phenomenon of false transparency, where surfaces intended to be opaque incorrectly appear transparent, remains largely unexplored in the literature. This oversight is significant because false transparency contributes substantially to view inconsistency artifacts through a different mechanism from popping.

**Object-centric reconstruction.** Scene reconstructions mostly reconstruct the front side of some objects. In contrast, object-centric techniques scan around the target object, and often have the object isolated from the scene. Some methods use object masks for targeted reconstruction [25, 26, 27], while others employ semantic segmentation or prompt-based interaction [28, 29, 30, 31]. These advancements have enabled object-aware representations, manipulation [32, 33], and improved 3D asset editing [34, 35]. However, accurately defining object boundaries in complex scenes with occlusions, transparencies, and intricate geometries remains challenging.

## 3   Theories & Methods

### 3.1   False-transparency mechanism

**Background.** In the original 3DGS pipeline [1], each pixel value is obtained by front-to-back $\alpha$-blending of the depth-sorted splats whose projection cover that pixel:

$$\mathbf{C}(\mathbf{p}) = \sum_{i \in \mathcal{N}(\mathbf{p})} \alpha_i \, \mathbf{c}_i \prod_{j<i} (1 - \alpha_j), \tag{1}$$

where $\alpha_i \in [0, 1]$ and $\mathbf{c}_i \in \mathbb{R}^3$ are the opacity and color of splat $i$ and $\mathcal{N}(\mathbf{p})$ is the front-to-back list along the ray going through pixel $\mathbf{p}$. Training minimizes a purely image-space objective with

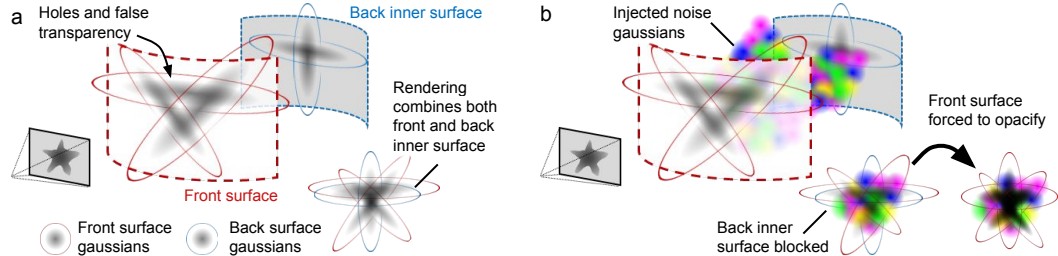

Figure 2: Mechanisms of false transparency and NGS. (a) A semi-transparent front surface and a inner back surface can jointly render to mimic a ground-truth image, causing the false transparency artifact. (b) NGS fills interior volume with colored noise Gaussians, promoting the surface opacity.

compound L1 and SSIM loss:

$$\mathcal{L}_{\text{photo}} \;=\; (1-\lambda)\,\frac{1}{|\mathcal{P}|}\sum_{\mathbf{p}\in\mathcal{P}}\big\|\mathbf{C}(\mathbf{p};\Theta) - \mathbf{I}_{\text{GT}}(\mathbf{p})\big\|_1 \;+\; \lambda\,\mathcal{L}_{\text{D-SSIM}}, \tag{2}$$

with respect to all Gaussian parameters $\Theta$. The photometric loss (Eq. (2)) constrains only the final RGB value and its local structure. It is indifferent to how the color is distributed along the ray.

**Where the transparency ambiguity arises.** Let $s$ be the index of the first intersected surface splat and denote all deeper splats by $\mathcal{B}$. Decomposing (1) at $s$ yields:

$$\mathbf{C}(\mathbf{p}) \;=\; \underbrace{\alpha_s\,\mathbf{c}_s}_{\text{front surface}} \;+\; \underbrace{(1-\alpha_s)\sum_{i\in\mathcal{B}}\alpha_i\,\mathbf{c}_i\prod_{s<j<i}\big(1-\alpha_j\big)}_{\text{background that leaks if } \alpha_s < 1}. \tag{3}$$

For every semi-transparent surface ($\alpha_s < 1$) there exists a set of background opacities/colors that reproduces the exact pixel color of a fully opaque surface, giving the same loss in Eq. (2). Gradient descent therefore can accept trivial minima with low $\alpha_s$. In turn, the surface becomes translucent even though it should be opaque (Fig. 2).

## 3.2 Object-centric reconstruction

**Alpha-consistency loss.** Given a pre-computed binary mask we can suppress any opacity that spills outside the object. Let $\mathcal{P} = \{1,\dots,h\,w\}$ be the set of pixel indices, $A_i \in [0,1]$ the rendered alpha and $M_i \in \{0,1\}$ the mask. Following [25] we define the **background-suppression loss** $\mathcal{L}_b = \frac{1}{|\mathcal{P}|}\sum_{i\in\mathcal{P}} A_i\big(1 - M_i\big)$, which drives $A_i \to 0$ wherever the mask is zero.

Most 3DGS variants employ strategies that restrict opacity to reduce over-reconstruction. For instance, the original 3DGS resets opacity values [1], Markov Chain Monte Carlo (MCMC) densifies the point cloud using sampling from Gaussian opacity distribution [14], and Revised Adaptive Density Control (ADC) implements explicit opacity regularization [12]. All of these approaches unintentionally promote transparency in scenarios where internal background colors match object surface colors, because there is no photometric incentive to maintain surface consistency when the optimization can minimize loss by simply reducing opacity. Adapting the Revised ADC idea to the object mask yields the complementary **foreground-opacity loss** $\mathcal{L}_f = \frac{1}{|\mathcal{P}|}\sum_{i\in\mathcal{P}}\big(1 - A_i\big)M_i$, which rewards pixels inside the mask for reaching full opacity. Summing the two gives a single alpha-consistency loss $\mathcal{L}_a$, which is simply the $L_1$ distance because $A_i, M_i \in [0,1]$:

$$\mathcal{L}_a \;=\; \mathcal{L}_f + \mathcal{L}_b \;=\; \frac{1}{|\mathcal{P}|}\sum_{i\in\mathcal{P}}|A_i - M_i|, \tag{4}$$

**Limitations in 360° captures.** For reconstructions from incomplete views of the object, $\mathcal{L}_a$ cleanly separates object and background. In full 360° scans each camera ray almost always intersects with multiple surfaces. Gaussians behind the first surface are 'background' for that ray, but lie inside the mask for other views. Without an additional bias toward the front-most Gaussians, the optimizer can still converge to semi-transparent surfaces.

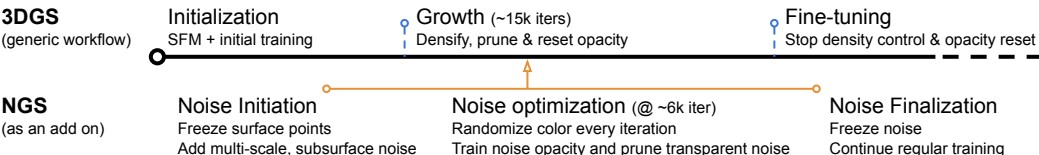

Figure 3: 3DGS training schedule with NGS. NGS is an add-on to the standard 3DGS pipeline.

## 3.3 Noise Guided Splatting (NGS)

NGS addresses false transparency by employing an alpha consistency loss and strategically placing noise Gaussians within the object's volume to obstruct direct lines of sight between front and back surfaces, as illustrated in Fig. 1, thereby forcing the optimization to prioritize an opaque foreground (Fig. 2). The pipeline begins by initializing a set of noise Gaussians within a coarse voxelized convex hull of the object. The color of the noise Gaussians are randomized in each iteration to prevent overfitting. These noise Gaussians are then pruned based on depth, ensuring only noise Gaussians inside the object remain. This initialization and pruning sequence is repeated in a multi-scale manner across increasing voxel resolutions to accurately fill complex geometries while remaining memory efficient. Afterwards, we conduct a brief fine tuning phase where the surface Gaussians are frozen, and the noise Gaussians opacities are trained and pruned. Finally, the noise Gaussians are frozen, the surface Gaussians are unfrozen and training proceeds normally with a reset learning rate.

**Initialization.** We first compute a convex hull mesh from the means of existing Gaussian primitives using Quickhull [36]. The mesh provides an approximation of the object's volume that is then converted into a coarse occupancy volume. Voxels located inside the mesh are marked as occupied. We map each occupied voxel in this coarse voxel grid to a sparse noise point, and set it to a random color. The dense occupancy grid is always synced to the sparse point in downstream operations.

**Pruning.** The convex hull approximation includes regions outside the actual object. Therefore, for each rendered pixel in each view in the capture, we identify and remove noise Gaussians that appear in front of surface Gaussians using 3DGS's depth ordering [1]. This carves away incorrectly placed noise Gaussians that would otherwise interfere with surface reconstruction. After depth pruning, we further eroded the occupancy grid to establish a buffer distance between the Gaussians of noise and the object surface. This guarantees a minimal thickness for the surface Gaussians to optimize without noise interference.

**Multi-scale noise injection.** For a balanced setup to minimize computational overhead and maximize the coverage of fine geometrical features, we initiate noise Gaussians at multiple-voxel grid resolution. The initialization is repeated in a coarse-to-fine manner, where we only initialize new noise Gaussians in occupied voxels that do not already contain noise Gaussians from previous resolution levels. This allows us to fit complex geometry and finer features while maintaining low noise count.

**Fine tuning.** As a final refinement step, we freeze the surface Gaussians and conduct a training pass where only the opacity values of noise Gaussians are made trainable. During this phase and for the remainder of training, we randomly set the color of each noise Gaussian from Red Green Blue Cyan Magenta Yellow (RGBCMY) at each iteration. This prevents the noise opacities from overfitting to the surface's color, ensuring they are optimized based on occlusion alone. RGBCMY is chosen because these six primary and secondary colors offer maximum contrast against most natural surface colors in RGB and HSV space. The opacity of incorrectly placed noise Gaussians (*e.g.,* too close to the surface) decreases until they are removed through pruning. The fine tuning is a final step to guarantee no noise Gaussians interact negatively with the surface Gaussians.

**Guided Surface training.** Following the fine-tuning phase, the interior noise structure is considered final. We freeze all parameters of the noise Gaussians for the remainder of training. The surface Gaussians are then unfrozen, and we reset the learning rate for their means. This reset allows the surface parameters to adapt effectively to the new optimization landscape defined by the internal noise barrier. Training then proceeds as normal, with the continued randomization of the noise Gaussians' colors preventing the surface from learning to complement a static internal pattern.

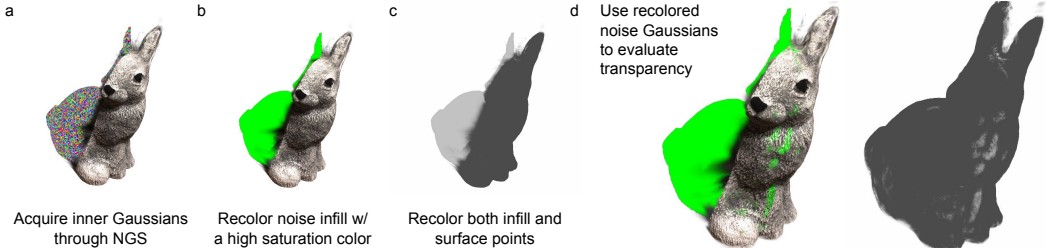

a     b     c     d   Use recolored noise Gaussians to evaluate transparency

Acquire inner Gaussians through NGS     Recolor noise infill w/ a high saturation color     Recolor both infill and surface points

Figure 4: Noise Gaussian primitives from NGS as an infill to evaluate surface transmittance. (a) Generated and extracted noise Gaussians from NGS training. (b) Recolored noise Gaussians for transparency visualization. (c) Recolored surface and noise Gaussians for higher visual contrast and quantitative analysis. (d) Noise Gaussians inserted into models trained with other methods to characterize false transparency.

### 3.4 Transparency benchmark

Standard quality metrics (*e.g.,* Peak Signal-to-Noise Ratio (PSNR), Structural Similarity Index (SSIM), and Learned Perceptual Image Patch Similarity (LPIPS)) do not quantify view inconsistency, because they are designed to work on pairs of static images. In scenarios where reconstructions exhibit false transparency, the metrics can still report high scores. Using noise Gaussian primitives created by NGS, we can create a dense point cloud placed inside the object volume that breaks the line of sight and obscures the background Gaussians during evaluation. It creates a false transparency diagnostic tool to visualize and measure the false transparency.

**Visualization.** For transparency visualization, we simply insert the pre-trained noise into a trained 3DGS asset and render them together using the rasterizer (Fig. 4). We can then visualize and measure the surface transparency in static renderings without interactively manipulating the 3D model to spot view inconsistency. The infill can be used to ravel transparency in the original model, or we could recolor the surface into a complementary color for better visual contrast (Fig. 4).

**Quantification.** To quantify the transparency, we set the infill and surface Gaussians into two separate channels (*e.g.,* green and red). We then render the image using the standard rasterizer, where the infill color channel in the render is the surface transmittance map $T_i$ for viewpoint $i$. The mean pixel values of the infill color channel are computed for all foreground pixels. If a segmentation mask $M_i$ is available, we apply the mask to further refine the target area. We could also use the rendering $\alpha$ or recolored surface Gaussian as $M_i$ when the segmentation mask is not available (*e.g.,* in novel views). We define results normalized in log scale as Surface Opacity Score ($SOS$):

$$SOS_i = \frac{log(\sum T_i/\sum M_i + \epsilon)}{log(\epsilon)}, \tag{5}$$

where $\epsilon$ is set to 1E-10 to ensure numerical stability. We expect $SOS_i = 1$ for fully opaque surfaces, and $SOS_i \approx 0$ for fully transparent surfaces.

## 4 Experiments & results

### 4.1 Experimental settings

**Dataset.** We used public available object-centric datasets, DTU [37] and OmniObject3D [38], to evaluate NGS. To support research on high-resolution macro 3D scanning, we also captured a novel high-resolution dataset, herein referred to as the *Stone Dataset*. This dataset comprises over 100 distinct stone samples, selected for their complex geometry and surface textures, to test the robustness and detail-capturing capabilities of our approach. For the *Stone Dataset*, we acquired 240 images per sample. Each stone was positioned on a rotating turntable within a softbox. Images were captured from 6 latitudinal and 40 longitudinal angles, covering the entire upper hemisphere of the samples. Images have $3000 \times 4000$ pixels, and 16-bit raw Bayer data was preserved to retain maximum image information. Camera pose estimation for this custom dataset was performed using COLMAP [39]. For all datasets employed in our study (DTU, OmniObject3D, and the *Stone Dataset*), we also produced a high-quality foreground segmentation for each image using MVAnet [40]. Example data

is shown in Fig. S1,S2&S3. All data created for this study will be made available to the community. This release includes the *Stone Dataset*, foreground segmentation masks, the generated noise infills, and a supplementary dataset featuring a mixture of everyday objects (Fig. S4).

**Training.** We conducted all experiments on NVIDIA L40S GPUs using the GSplat framework [41]. Unless explicitly stated, our base implementation of NGS used the default variant of Gsplat. Trainings typically required no more than 8GB of VRAM. Average training time for base methods (without noise) was 16 min for DTU [37], 11 min for OmniObject3D [38], and 18 min for the new Stone dataset (15k adaptive density control + 15k refinement). Adding noise Gaussians increased memory consumption by about 50% due to the additional primitives required to fill the object interior, and increased training time by 1 min. If the memory consumption becomes a limiting factor, reducing the number of finer noise Gaussians can substantially reduce the memory consumption.

**Baselines.** We conducted our transparency evaluation protocol (Sect. 3.4) on several 3DGS variants, Gaussian Opacity Fields (GOF) [42], and StopThePop [6], across all three datasets (DTU, OmniObject3D, and our *Stone dataset*)

**Benchmarking.** A standard 7:1 train-test split was used for all datasets, with the test set forming the basis for all quantitative metrics. We first assessed the quality of our result and baselines using PSNR, SSIM LPIPS and our proposed $SOS$ metric. To analyze the perceptual impact of any surface transparency, the standard NVS metrics were re-evaluated with pre-trained, recolored noise infill included in the rendered scene. Comparing these infill-conditioned metrics (denoted with an asterisk in Table. 1, e.g., PSNR*) against the baseline NVS scores highlights the extent to which the infill visually "leaks" through the object's surface.

## 4.2 Implementation details

**NGS settings.** The $\mathcal{L}_a$ loss was added to the photometric loss from $\mathcal{L}_{photo}$ to enforce $\alpha$-consistency. Noise Gaussians were introduced at iteration 6,000 during adaptive density control, allowing sufficient time for the initial surface reconstruction to establish before applying our transparency guidance strategy (Fig. 3). We refined the noise Gaussians for 1,000 iterations. The surface Gaussians were frozen during noise refinement. After the noise refinement, the noise Gaussians' means, opacity, scale and rotations were frozen. Until the end of training, each noise Gaussian's color was randomized from RGBCMY at each iteration, preventing the surface Gaussians from fitting a fixed noise pattern. We reset the learning rate of Gaussian means to compensate for the sudden change to the blending. The rest of training follows the default GSplat parameters.

## 4.3 Object-centric 3DGS Reconstruction Evaluation

**Novel view synthesis metrics and limitations.** NVS metrics (PSNR, SSIM, LPIPS) effectively measure the overall visual fidelity of reconstructions but fail to specifically identify or quantify false transparency artifacts. These artifacts stem from their foreground-background ambiguity, where background elements visible through transparent surfaces contribute to the render score. This is particularly problematic for object-centric reconstruction where internal Gaussians can be optimized to match the appearance of surface Gaussians. To address this limitation, we employ our transparency evaluation protocol described in Section 3.4.

**Quantitative results.** Our quantitative evaluation (Table 1) demonstrates that NGS consistently improves surface opacity across all tested methods while maintaining or slightly improving standard rendering metrics. When comparing transparency scores between baseline methods and their noise-enhanced counterparts, we observe an average reduction in surface transmittance by a factor of two, confirming the effectiveness of our approach in addressing false transparency. Notably, the introduction of noise Gaussians does not adversely affect rendering quality as measured by standard metrics, with several cases showing modest improvements in PSNR and SSIM. This suggests that by resolving the foreground-background optimization ambiguity, NGS not only reduces transparency but also helps the optimization process converge to more accurate reconstructions.

**Quality comparison on challenging cases.** Our evaluation demonstrates varying degrees of effectiveness across different datasets. On our *Stone dataset*, NGS almost completely eliminates false transparency artifacts (Fig. 5), resulting in fully opaque surface reconstructions. This superior performance can be attributed to the controlled capture environment with consistent, diffuse light

Table 1: Average NVS and $SOS$ results on DTU (Table. S2), OmniObject3D (Table. S3) and our novel dataset (Table. S1). Metrics denoted by * were acquired when using a green infill, as shown in Fig. 5 and $+\alpha$ denotes use of $\mathcal{L}_a$.

| | Method | PSNR↑ | PSNR*↑ | SSIM↑ | SSIM*↑ | LPIPS↓ | LPIPS*↓ | $SOS$↑ |
|---|---|---|---|---|---|---|---|---|
| | 3DGS | 25.575 | 22.967 | **0.891** | 0.874 | **0.180** | 0.250 | 0.147 |
| | GOF | **25.648** | 21.109 | 0.880 | 0.816 | 0.209 | 0.273 | 0.179 |
| DTU | StopThePop | 22.817 | 18.885 | 0.852 | 0.780 | 0.213 | 0.315 | 0.135 |
| | GSplat+$\alpha$ | 25.435 | 25.263 | 0.884 | **0.883** | 0.183 | **0.186** | 0.598 |
| | NGS | 25.428 | **25.427** | 0.881 | 0.881 | 0.192 | 0.192 | **0.749** |
| | 3DGS | **34.610** | 27.551 | 0.949 | 0.909 | 0.055 | 0.222 | 0.140 |
| | GOF | 31.469 | 21.998 | 0.893 | 0.780 | 0.186 | 0.324 | 0.126 |
| Stone | StopThePop | 32.457 | 23.047 | 0.945 | 0.853 | 0.078 | 0.223 | 0.168 |
| | GSplat+$\alpha$ | 33.832 | 33.823 | 0.948 | 0.948 | 0.062 | 0.062 | 0.891 |
| | NGS | 34.148 | **34.148** | **0.951** | **0.951** | **0.053** | **0.053** | **0.922** |
| | 3DGS | 29.300 | 27.456 | 0.940 | 0.929 | 0.069 | 0.116 | 0.215 |
| | GOF | 32.259 | 24.931 | 0.970 | 0.898 | 0.062 | 0.122 | 0.208 |
| OmniObject | StopThePop | 32.274 | 25.095 | 0.970 | 0.900 | **0.050** | 0.113 | 0.265 |
| | GSplat+$\alpha$ | 33.575 | 33.350 | **0.973** | **0.972** | 0.060 | 0.064 | 0.642 |
| | NGS | **33.619** | **33.578** | 0.972 | **0.972** | 0.060 | **0.060** | **0.736** |

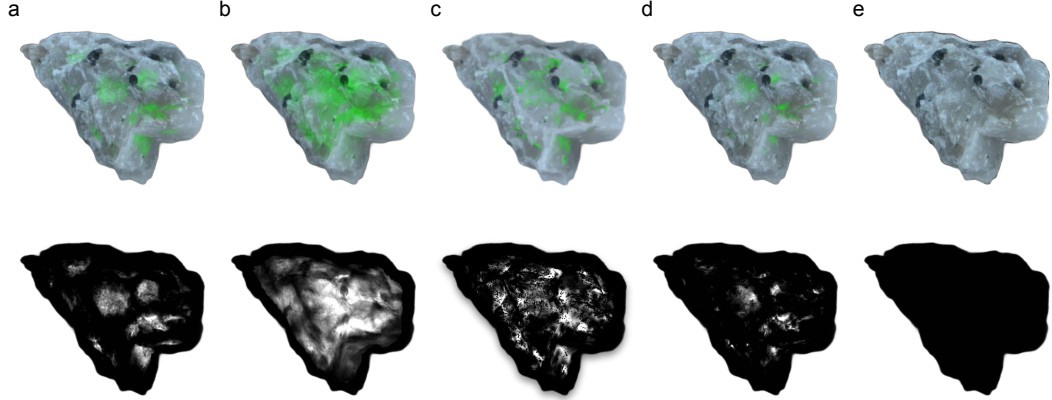

Figure 5: Renders with green infill revealing transparency (top) and corresponding transmittance maps (bottom) for (a) 3DGS, (b) GOF, (c) StopthePop, (d) Gsplat+$\alpha$ and (e) NGS.

from a softbox, which reduces view-dependent effects and provides uniform illumination across all viewpoints. In contrast, while still showing significant improvement, NGS achieves more modest transparency reduction on the DTU and OmniObject3D datasets (Fig. S5, S6, S7 & S8 ). These datasets feature more variable lighting conditions with stronger directional components, creating view-dependent effects that can be misinterpreted as transparency during optimization. The inconsistent shadows and highlights across different viewpoints make it more challenging to establish a clear distinction between surface appearance variation and actual transparency. These results highlight the important relationship between lighting consistency and false transparency in 3DGS, suggesting that controlled capture conditions can substantially enhance the effectiveness of transparency reduction.

## 4.4 Ablation Study

To thoroughly evaluate the impact of each component in our NGS method, we conducted a series of ablation experiments using our stone dataset to isolate the effects of individual design choices on both rendering quality and transparency reduction. One at a time, we removed the following

Table 2: Mean NVS and $SOS$ scores over the *Stone Dataset*. Metrics denoted by * were acquired when using an infill. The current NGS setting provides a good balance between $SOS$ and NVS metrics. Renderings of a selected stone is shown in Fig. S9

| Method | PSNR↑ | PSNR*↑ | SSIM↑ | SSIM*↑ | LPIPS↓ | LPIPS*↓ | $SOS$↑ |
|---|---|---|---|---|---|---|---|
| Ours | **32.201** | **32.201** | **0.934** | **0.934** | 0.145 | 0.145 | 0.969 |
| w/o Erosion | 30.785 | 30.785 | 0.914 | 0.914 | 0.225 | 0.225 | **1.000** |
| w/o Pruning | 31.496 | 31.489 | 0.927 | 0.927 | 0.151 | 0.151 | 0.952 |
| w/o $\mathcal{L}_f$ | 31.252 | 30.930 | 0.928 | 0.927 | **0.126** | **0.135** | 0.379 |
| w/o LR reset | 26.136 | 26.133 | 0.830 | 0.830 | 0.416 | 0.416 | 0.545 |
| Random Bg | 30.205 | 30.149 | 0.923 | 0.923 | 0.136 | 0.139 | 0.467 |
| w/o Color reset | 31.442 | 31.436 | 0.926 | 0.926 | 0.154 | 0.154 | 0.962 |

components from the proposed NGS pipeline: binary erosion, pruning, $\mathcal{L}_f$ (from Eq. 4) and learning rate reset at noise initialization. We also substitute $\mathcal{L}_f$ for random background. Our results (Table 2) demonstrate that the default NGS configuration achieves an optimal balance between rendering quality and transparency reduction. Among the tested variations, two components proved particularly crucial for the method's effectiveness, learning rate reset and $\mathcal{L}_f$.

**Learning rate reset.** Resetting the learning rate decay for Gaussian means after noise introduction substantially improved the system's ability to adapt to the new optimization landscape. Without this reset, we observed that surface Gaussians struggled to properly adjust their parameters in response to the presence of internal noise Gaussians, resulting in degraded reconstruction quality and higher transparency scores. $\mathcal{L}_f$ outperformed alternative background regularization techniques. While random background approaches provided some mitigation of transparency issues, our foreground loss directly incentivizes surface opacity, resulting in a greater reduction in transparency scores.

**Noise voxel grid erosion.** We also note that although not eroding the voxel grid for noise Gaussians achieves better transparency scores, it hinders the visual quality of the renders.

This study confirms that NGS's effectiveness stems from the synergistic combination of appropriate noise guidance and targeted optimization adjustments, with the learning rate reset and foreground loss serving as the most significant contributors to its performance.

## 5 Discussion

**Findings.** Our results demonstrate that Noise Guided Splatting effectively addresses the false transparency problem in object-centric 3D Gaussian Splatting while maintaining or improving NVS quality. By injecting noise Gaussians within object volumes, we successfully force the optimization process to prioritize surface opacity, resulting in more accurate and view-consistent reconstructions. The results align with our theoretical understanding of the transparency problem as an optimization ambiguity between foreground and background elements.

Standard photometric losses alone cannot distinguish between a properly opaque surface and a partially transparent surface with a solid surface sitting behind. By breaking the line of sight between front and back surfaces with noise infill, we eliminate this ambiguity and guide the optimization toward solutions with appropriate surface opacity. The reduction in $SOS$ across different methods and datasets confirms the generalizability of our approach. Importantly, this improvement does not come at the cost of rendering quality and even improves PSNR and SSIM in some cases. It demonstrates that resolving false transparency is not merely a visual enhancement but a fundamental improvement to the reconstruction process.

**Limitations.** Despite its effectiveness, NGS has several limitations worth acknowledging. First, our method benefits from high-quality segmentation masks to guide the object-centric reconstruction. Inaccurate segmentation can lead to incorrect noise Gaussian placement, potentially compromising reconstruction quality. This limitation is particularly relevant for objects with fine details or complex boundaries. Second, NGS assumes the material is fully opaque. For inherently transparent or

translucent objects (e.g., glass, certain plastics), our approach may incorrectly enforce opacity where transparency is actually desired. Third, our approach requires a reasonable Gaussian point cloud to define a convex hull encompassing the entirety of the object. Fourth, while we have shown NGS's effectiveness for object-centric reconstruction, its application to large-scale scene reconstruction requires further investigation. The current noise initialization and pruning strategies will require adaptation for scenes with multiple objects and complex spatial relationships. Finally, this method is less effective for thin structures where injecting noise is difficult.

**Computational Considerations.** NGS introduces some computational overhead relative to standard 3DGS, primarily in the form of increased memory requirements and rendering time during training. The number of additional noise Gaussians depends on object geometry complexity, typically increasing the total Gaussian count by 30-50%. This translates to proportional increases in memory usage and rendering time. However, several optimizations could mitigate these costs. Our multi-resolution initialization approach already reduces the number of required noise Gaussians compared to uniform voxel filling. Further improvements include reducing the parameters for noise Gaussians down to spheres with variable opacity and color, removing spherical harmonics, quaternions and scales.

**Broader Applications.** The improved surface consistency provided by NGS has significant implications for several downstream applications. In AR/VR asset creation, where accurate object representation is critical for immersive experiences, our method produces more reliable reconstructions with well-defined boundaries. The reduced transparency particularly benefits applications requiring watertight meshes, such as physics simulations or 3D printing. By providing clearer surface definitions, NGS reduces the need for manual cleanup of extracted meshes, streamlining the transition from digital reconstruction to physical reproduction. It is also worth noting that while NGS effectively addresses the false transparency problem, future research may develop alternative approaches to resolve this issue. Nevertheless, the infill assets generated by NGS can still serve a valuable purpose. They provide a robust basis for the $SOS$ benchmark, thereby facilitating the evaluation and comparison of subsequent methods aimed at mitigating false transparency, regardless of their underlying technique.

**Future Directions.** A logical extension of this work involves applying it to more complex scenes containing multiple objects using segmentation methods like Segment Any 3D Gaussians [43] to isolate each object and apply our NGS pipeline individually. Another promising research direction to emerge from this work is to use the noise infill as learnable interior Gaussians and predict the internal structures of the model. For example, FruitNinja [44] used infill particles together with a diffusion model to generate the internal structures of 3D models. The noise infill provide a more robust way for initializing these method.

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

# A    Appendix / Acronyms

**3DGS**  3D Gaussian Splatting

**NGS**  Noise Guided Splatting

**PSNR**  Peak Signal-to-Noise Ratio

**SSIM**  Structural Similarity Index

**LPIPS**  Learned Perceptual Image Patch Similarity

**RGBCMY**  Red Green Blue Cyan Magenta Yellow

**NeRF**  Neural Radiance Fields

**NVS**  Novel View Synthesis

$SOS$  Surface Opacity Score

**MCMC**  Markov Chain Monte Carlo

**GOF**  Gaussian Opacity Fields

**ADC**  Adaptive Density Control

**IQA**  Image Quality Assessment

# B   Appendix / Evaluation Results

Table S1: Full NVS and *SOS* results on the *Stone Dataset*. Metrics denoted by * were acquired when using an infill, as shown in Fig. 5 and +$\alpha$ denotes use of $\mathcal{L}_a$, and N° Gaussians denotes the number of Gaussians.

| Index | Method | PSNR | PSNR* | SSIM | SSIM* | LPIPS | LPIPS* | *SOS* | N° Gaussians |
|---|---|---|---|---|---|---|---|---|---|
| 1 | NGS | 32.563 | 32.563 | 0.947 | 0.947 | 0.062 | 0.062 | 1.000 | 585856 |
| | Gsplat+$\alpha$ | 32.260 | 32.260 | 0.943 | 0.943 | 0.069 | 0.069 | 1.000 | 551437 |
| 2 | NGS | 36.229 | 36.229 | 0.956 | 0.956 | 0.043 | 0.043 | 1.000 | 575300 |
| | Gsplat+$\alpha$ | 35.968 | 35.968 | 0.954 | 0.954 | 0.050 | 0.050 | 1.000 | 479522 |
| 3 | NGS | 33.418 | 33.418 | 0.945 | 0.945 | 0.071 | 0.071 | 0.926 | 591907 |
| | Gsplat+$\alpha$ | 33.168 | 33.168 | 0.942 | 0.942 | 0.081 | 0.081 | 0.835 | 476246 |
| 4 | NGS | 36.244 | 36.244 | 0.962 | 0.962 | 0.046 | 0.046 | 0.953 | 554600 |
| | Gsplat+$\alpha$ | 35.814 | 35.814 | 0.957 | 0.957 | 0.055 | 0.055 | 0.949 | 453943 |
| 5 | NGS | 32.871 | 32.871 | 0.953 | 0.953 | 0.053 | 0.053 | 0.958 | 542696 |
| | Gsplat+$\alpha$ | 32.703 | 32.703 | 0.951 | 0.951 | 0.061 | 0.061 | 0.927 | 442430 |
| 6 | NGS | 33.302 | 33.302 | 0.940 | 0.940 | 0.057 | 0.057 | 0.626 | 615372 |
| | Gsplat+$\alpha$ | 32.996 | 32.992 | 0.937 | 0.937 | 0.062 | 0.063 | 0.495 | 484779 |
| 7 | NGS | 32.993 | 32.993 | 0.947 | 0.947 | 0.053 | 0.053 | 0.913 | 540734 |
| | Gsplat+$\alpha$ | 32.785 | 32.785 | 0.944 | 0.944 | 0.061 | 0.061 | 0.912 | 423754 |
| 8 | NGS | 33.153 | 33.153 | 0.948 | 0.948 | 0.042 | 0.042 | 0.995 | 571105 |
| | Gsplat+$\alpha$ | 32.826 | 32.826 | 0.945 | 0.945 | 0.049 | 0.049 | 0.975 | 428108 |
| 9 | NGS | 36.517 | 36.517 | 0.960 | 0.960 | 0.048 | 0.048 | 0.961 | 585541 |
| | Gsplat+$\alpha$ | 35.973 | 35.972 | 0.956 | 0.956 | 0.058 | 0.058 | 0.950 | 502017 |
| 10 | NGS | 34.193 | 34.186 | 0.951 | 0.951 | 0.057 | 0.057 | 0.890 | 564590 |
| | Gsplat+$\alpha$ | 33.823 | 33.739 | 0.947 | 0.947 | 0.070 | 0.071 | 0.872 | 435205 |
| Mean | NGS | 34.148 | 34.148 | 0.951 | 0.951 | 0.053 | 0.053 | 0.922 | 572770 |
| | Gsplat+$\alpha$ | 33.832 | 33.823 | 0.948 | 0.948 | 0.062 | 0.062 | 0.891 | 467744 |
| Std. | NGS | 1.565 | 1.565 | 0.007 | 0.007 | 0.009 | 0.009 | 0.111 | 23220 |
| | Gsplat+$\alpha$ | 1.493 | 1.494 | 0.007 | 0.007 | 0.010 | 0.010 | 0.149 | 39548 |

Scan indices correspond to the following files: (1) scan_20250416_093245, (2) scan_20250416_140345, (3) scan_20250416_143850, (4) scan_20250416_151813, (5) scan_20250416_161137, (6) scan_20250416_165925, (7) scan_20250417_101115, (8) scan_20250417_112354, (9) scan_20250417_150930, (10) scan_20250417_153612

Table S2: Full NVS and $SOS$ results on DTU. Metrics denoted by * were acquired when using a green infill, as shown in Fig. 5 and $+\alpha$ denotes use of $\mathcal{L}_a$, and N° Gaussians denotes the number of Gaussians.

| | Method | PSNR | PSNR* | SSIM | SSIM* | LPIPS | LPIPS* | $SOS$ | N° Gaussians |
|---|---|---|---|---|---|---|---|---|---|
| 1 | NGS | 24.359 | 24.359 | 0.903 | 0.903 | 0.097 | 0.097 | 0.888 | 1461500 |
| | Gsplat+$\alpha$ | 24.372 | 24.365 | 0.905 | 0.905 | 0.092 | 0.093 | 0.773 | 2113100 |
| 2 | NGS | 22.896 | 22.894 | 0.888 | 0.888 | 0.227 | 0.227 | 0.585 | 404102 |
| | Gsplat+$\alpha$ | 22.787 | 22.283 | 0.890 | 0.887 | 0.220 | 0.229 | 0.418 | 566632 |
| 3 | NGS | 24.450 | 24.450 | 0.880 | 0.880 | 0.234 | 0.234 | 0.475 | 495216 |
| | Gsplat+$\alpha$ | 24.390 | 24.280 | 0.880 | 0.880 | 0.228 | 0.231 | 0.318 | 692157 |
| 4 | NGS | 27.541 | 27.541 | 0.905 | 0.905 | 0.148 | 0.148 | 0.953 | 782621 |
| | Gsplat+$\alpha$ | 27.679 | 27.679 | 0.909 | 0.909 | 0.139 | 0.139 | 0.978 | 1249151 |
| 5 | NGS | 25.682 | 25.682 | 0.877 | 0.877 | 0.196 | 0.196 | 0.791 | 567251 |
| | Gsplat+$\alpha$ | 25.609 | 25.515 | 0.878 | 0.878 | 0.187 | 0.189 | 0.367 | 859701 |
| 6 | NGS | 28.532 | 28.529 | 0.894 | 0.894 | 0.187 | 0.188 | 0.552 | 642402 |
| | Gsplat+$\alpha$ | 28.525 | 28.036 | 0.897 | 0.894 | 0.177 | 0.185 | 0.443 | 1007084 |
| 7 | NGS | 24.532 | 24.532 | 0.821 | 0.821 | 0.253 | 0.253 | 0.999 | 633185 |
| | Gsplat+$\alpha$ | 24.681 | 24.681 | 0.827 | 0.827 | 0.238 | 0.238 | 0.890 | 839702 |
| Mean | NGS | 25.428 | 25.427 | 0.881 | 0.881 | 0.192 | 0.192 | 0.749 | 712325 |
| | Gsplat+$\alpha$ | 25.435 | 25.263 | 0.884 | 0.883 | 0.183 | 0.186 | 0.598 | 1046790 |
| Std. | NGS | 1.978 | 1.978 | 0.029 | 0.029 | 0.054 | 0.054 | 0.211 | 351317 |
| | Gsplat+$\alpha$ | 2.017 | 2.024 | 0.027 | 0.027 | 0.053 | 0.054 | 0.273 | 518612 |

Scan indices correspond to the following files: (1) scan55, (2) scan65, (3) scan69, (4) scan106, (5) scan114, (6) scan118, (7) scan122

Table S3: Full NVS and $SOS$ results on OmniObject3D. Metrics denoted by * were acquired when using a green infill, as shown in Fig. 5 and $+\alpha$ denotes use of $\mathcal{L}_a$, and N° Gaussians denotes the number of Gaussians.

|  | Method | PSNR | PSNR* | SSIM | SSIM* | LPIPS | LPIPS* | $SOS$ | N° Gaussians |
|---|---|---|---|---|---|---|---|---|---|
| 1 | NGS | 35.848 | 35.608 | 0.975 | 0.975 | 0.098 | 0.100 | 0.555 | 191628 |
|  | Gsplat+$\alpha$ | 35.825 | 35.157 | 0.976 | 0.975 | 0.095 | 0.105 | 0.402 | 250896 |
| 2 | NGS | 31.690 | 31.690 | 0.978 | 0.978 | 0.039 | 0.039 | 0.806 | 223903 |
|  | Gsplat+$\alpha$ | 31.637 | 31.598 | 0.978 | 0.978 | 0.040 | 0.041 | 0.605 | 279212 |
| 3 | NGS | 35.682 | 35.671 | 0.980 | 0.980 | 0.020 | 0.020 | 0.817 | 123827 |
|  | Gsplat+$\alpha$ | 35.673 | 35.663 | 0.981 | 0.981 | 0.020 | 0.020 | 0.837 | 169815 |
| 4 | NGS | 37.958 | 37.958 | 0.972 | 0.972 | 0.083 | 0.083 | 1.000 | 231004 |
|  | Gsplat+$\alpha$ | 37.894 | 37.894 | 0.972 | 0.972 | 0.082 | 0.082 | 1.000 | 300919 |
| 5 | NGS | 33.284 | 33.283 | 0.977 | 0.977 | 0.045 | 0.045 | 0.439 | 274345 |
|  | Gsplat+$\alpha$ | 33.212 | 32.431 | 0.978 | 0.976 | 0.044 | 0.055 | 0.324 | 343927 |
| 6 | NGS | 32.178 | 32.178 | 0.972 | 0.972 | 0.048 | 0.048 | 0.834 | 223486 |
|  | Gsplat+$\alpha$ | 32.124 | 32.124 | 0.972 | 0.972 | 0.048 | 0.048 | 0.825 | 272385 |
| 7 | NGS | 28.693 | 28.658 | 0.953 | 0.953 | 0.085 | 0.088 | 0.699 | 268095 |
|  | Gsplat+$\alpha$ | 28.663 | 28.583 | 0.954 | 0.954 | 0.089 | 0.096 | 0.504 | 332044 |
| Mean | NGS | 33.619 | 33.578 | 0.972 | 0.972 | 0.060 | 0.060 | 0.736 | 219469.714 |
|  | Gsplat+$\alpha$ | 33.575 | 33.350 | 0.973 | 0.972 | 0.060 | 0.064 | 0.642 | 278456.857 |
| Std. | NGS | 3.115 | 3.096 | 0.009 | 0.009 | 0.029 | 0.030 | 0.188 | 50772.745 |
|  | Gsplat+$\alpha$ | 3.115 | 3.093 | 0.009 | 0.009 | 0.029 | 0.031 | 0.251 | 58112.765 |

Scan indices correspond to the following files: (1) antique_004, (2) dinosaur_004, (3) dinosaur_005, (4) antique_005, (5) dinosaur_006, (6) dinosaur_007, (7) dinosaur_008

# C Appendix / Supplementary Figures

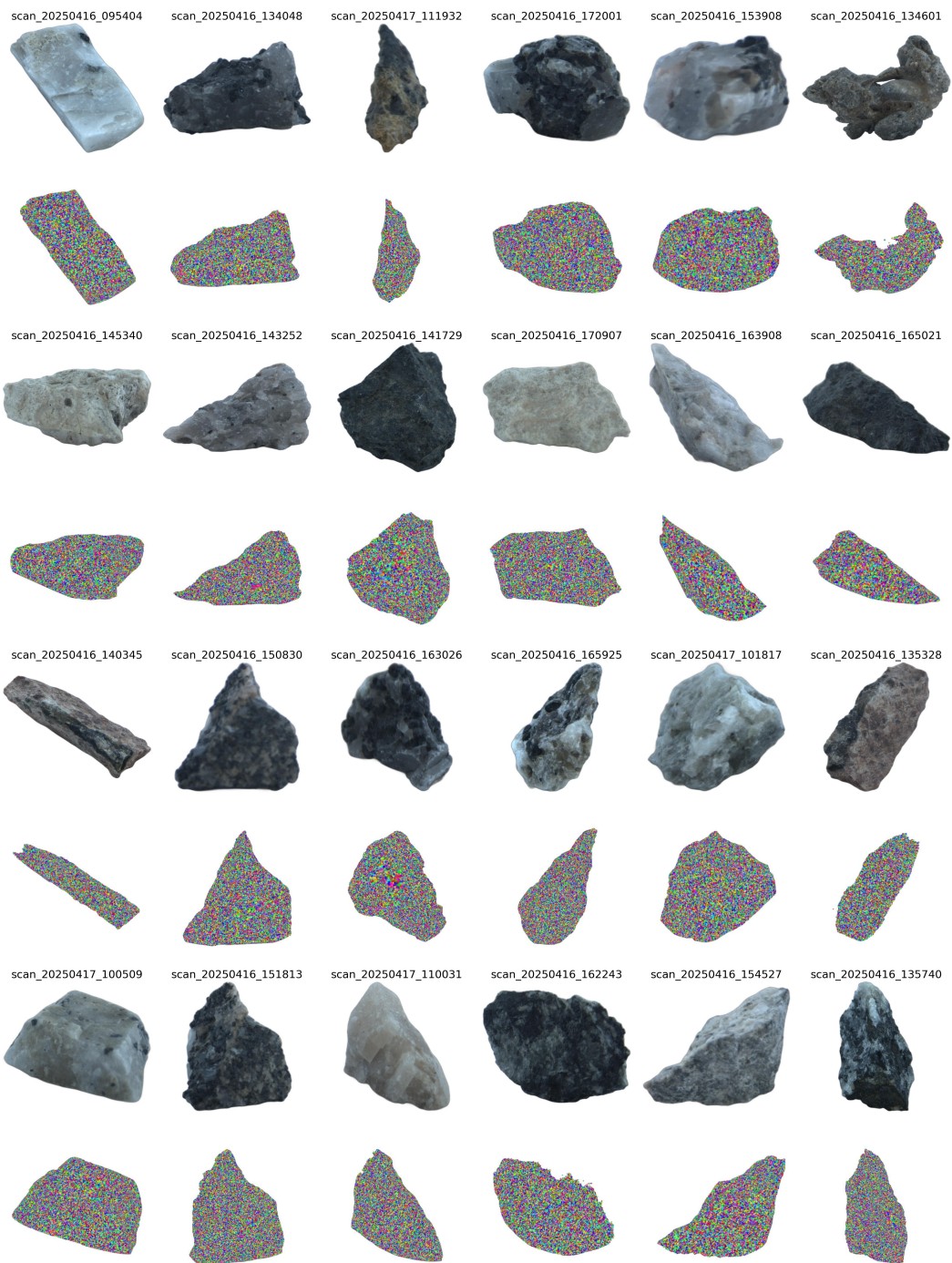

Figure S1: Stone dataset with noise augmentations

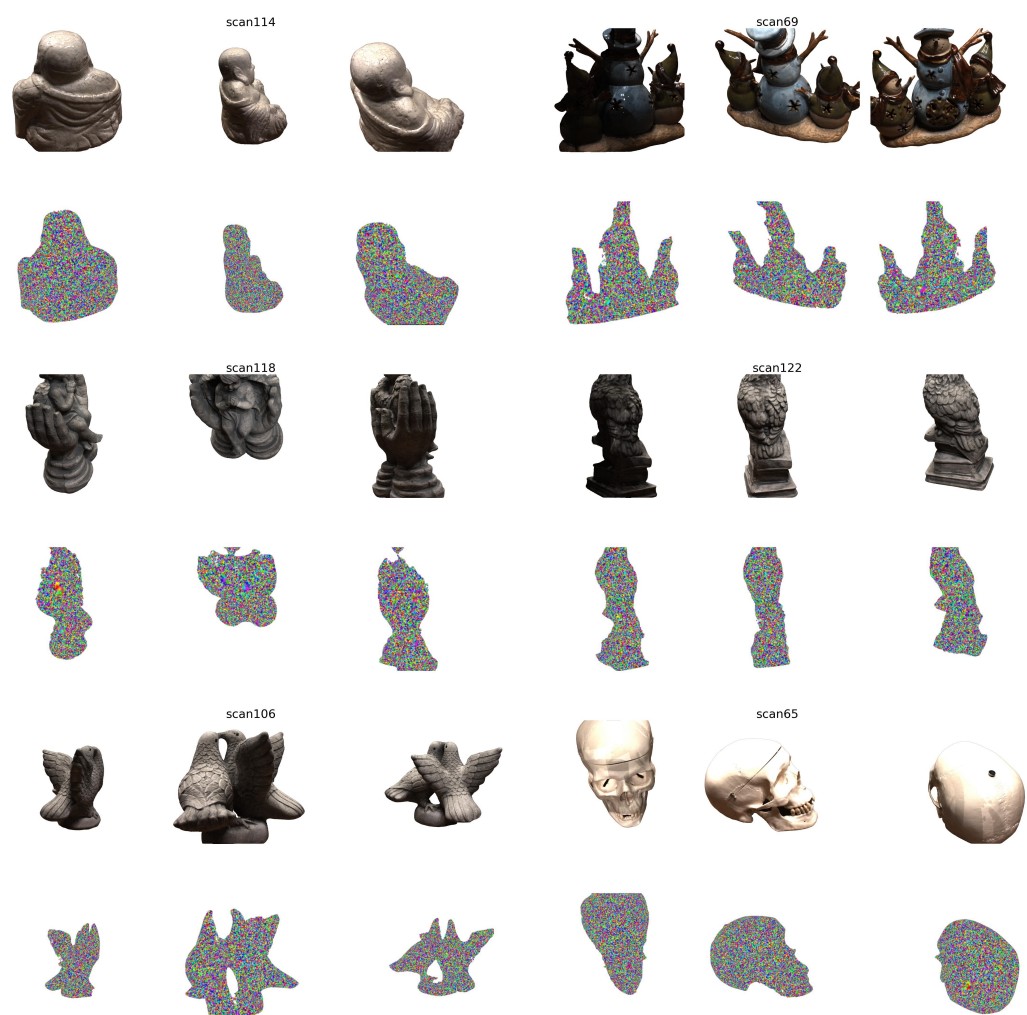

Figure S2: DTU with noise augmentation

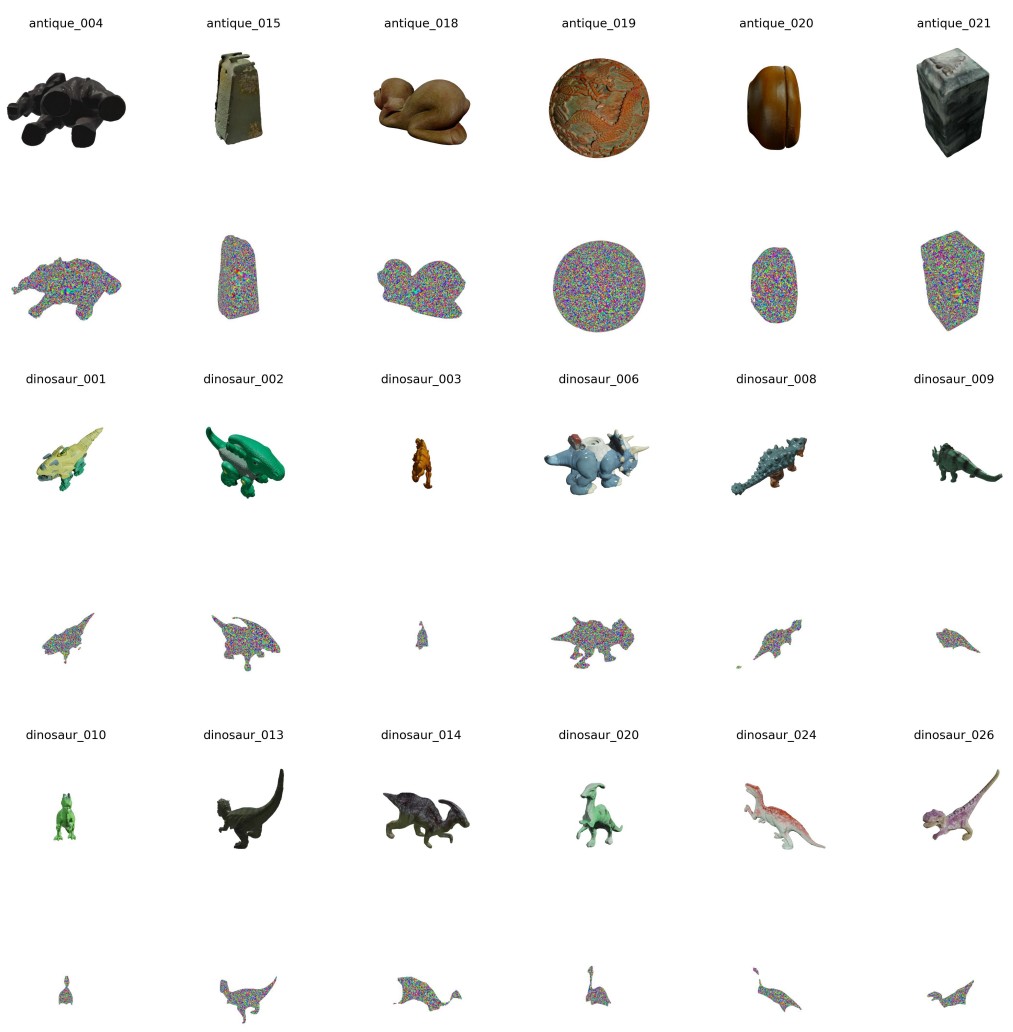

Figure S3: OmniObject3D with noise augmentation

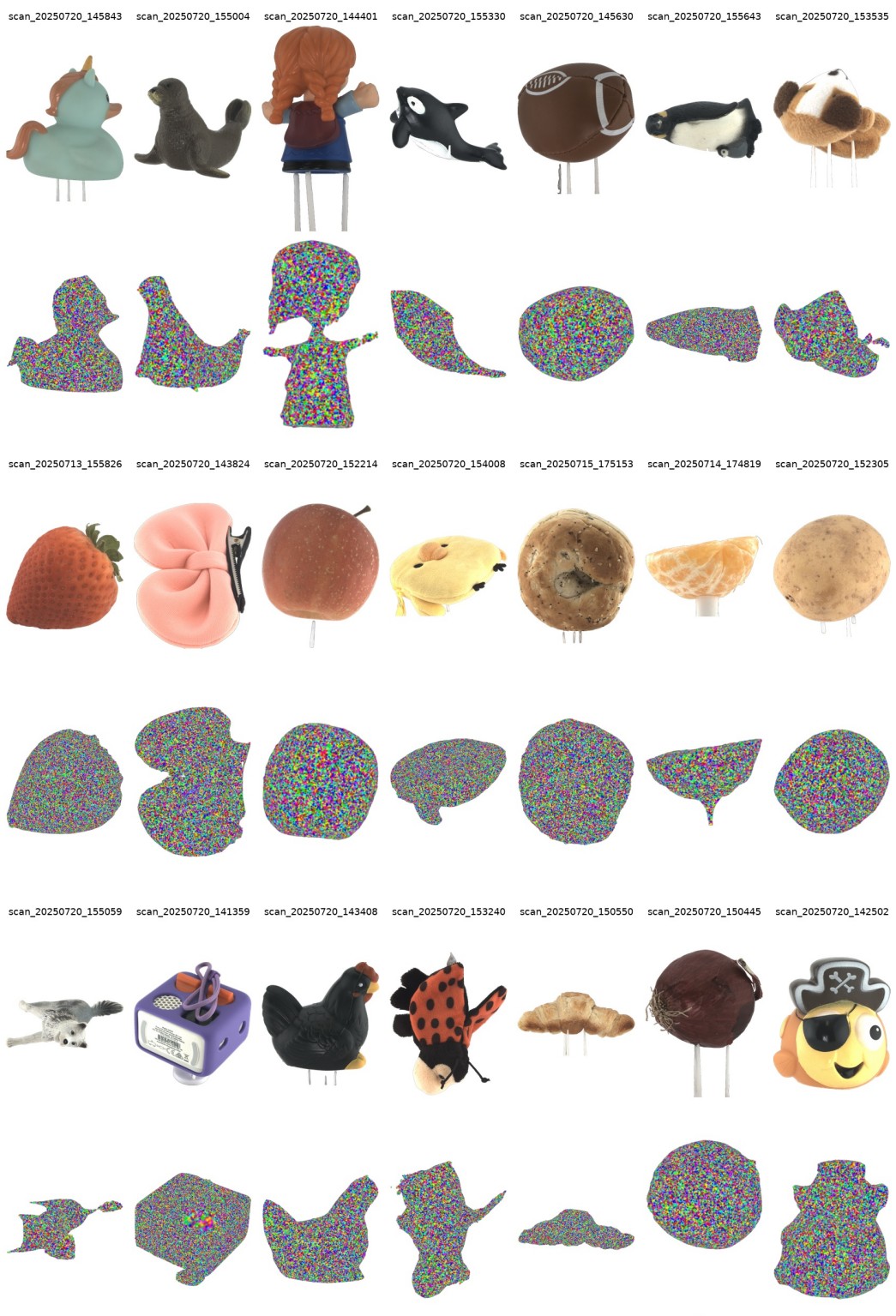

Figure S4: Object dataset with noise augmentation

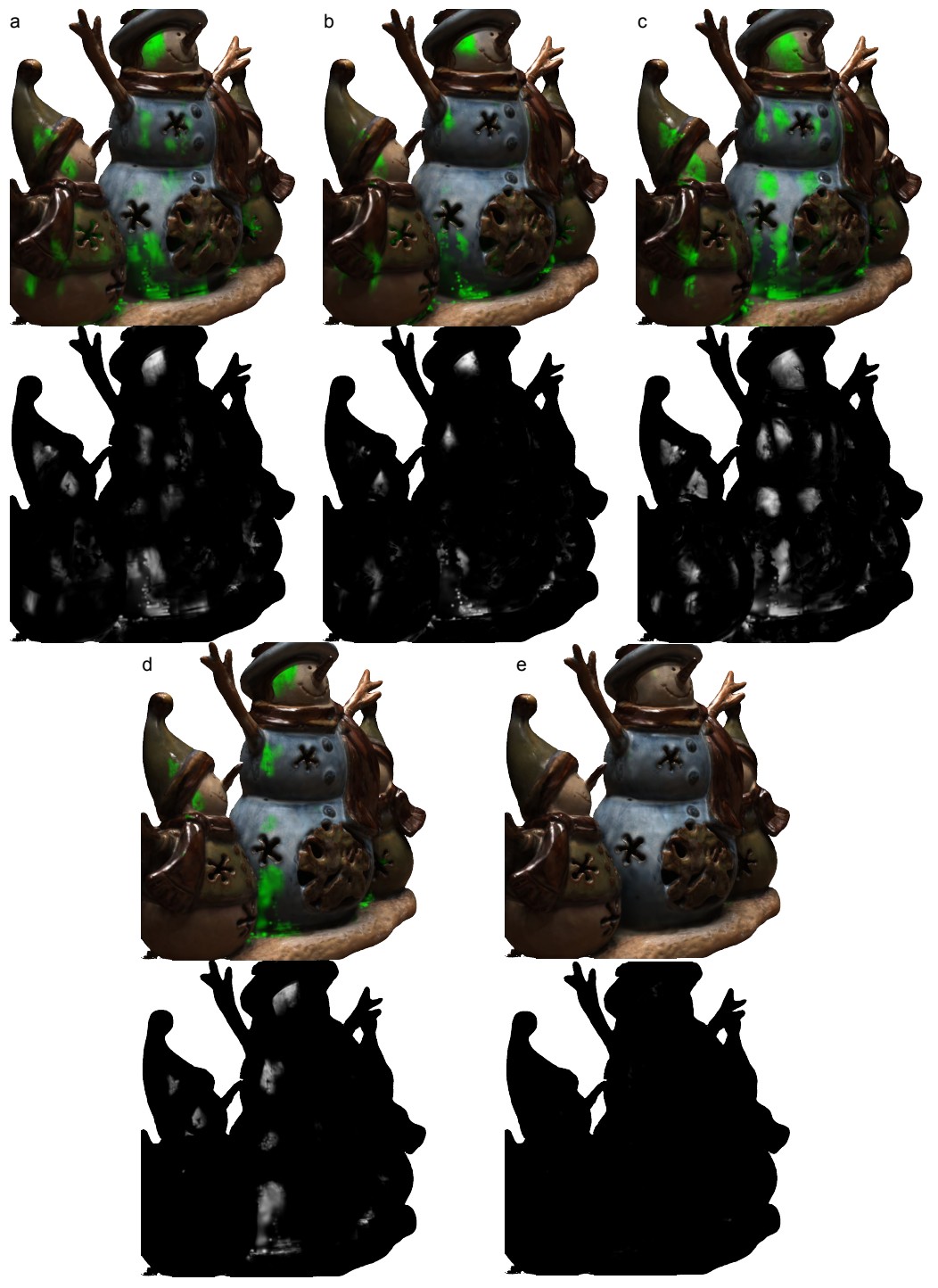

Figure S5: DTU renders with green infill revealing transparency (top) and corresponding transmittance maps (bottom) for (a) 3DGS, (b) GOF, (c) StopthePop, (d) Gsplat+$\alpha$ and (e) NGS.

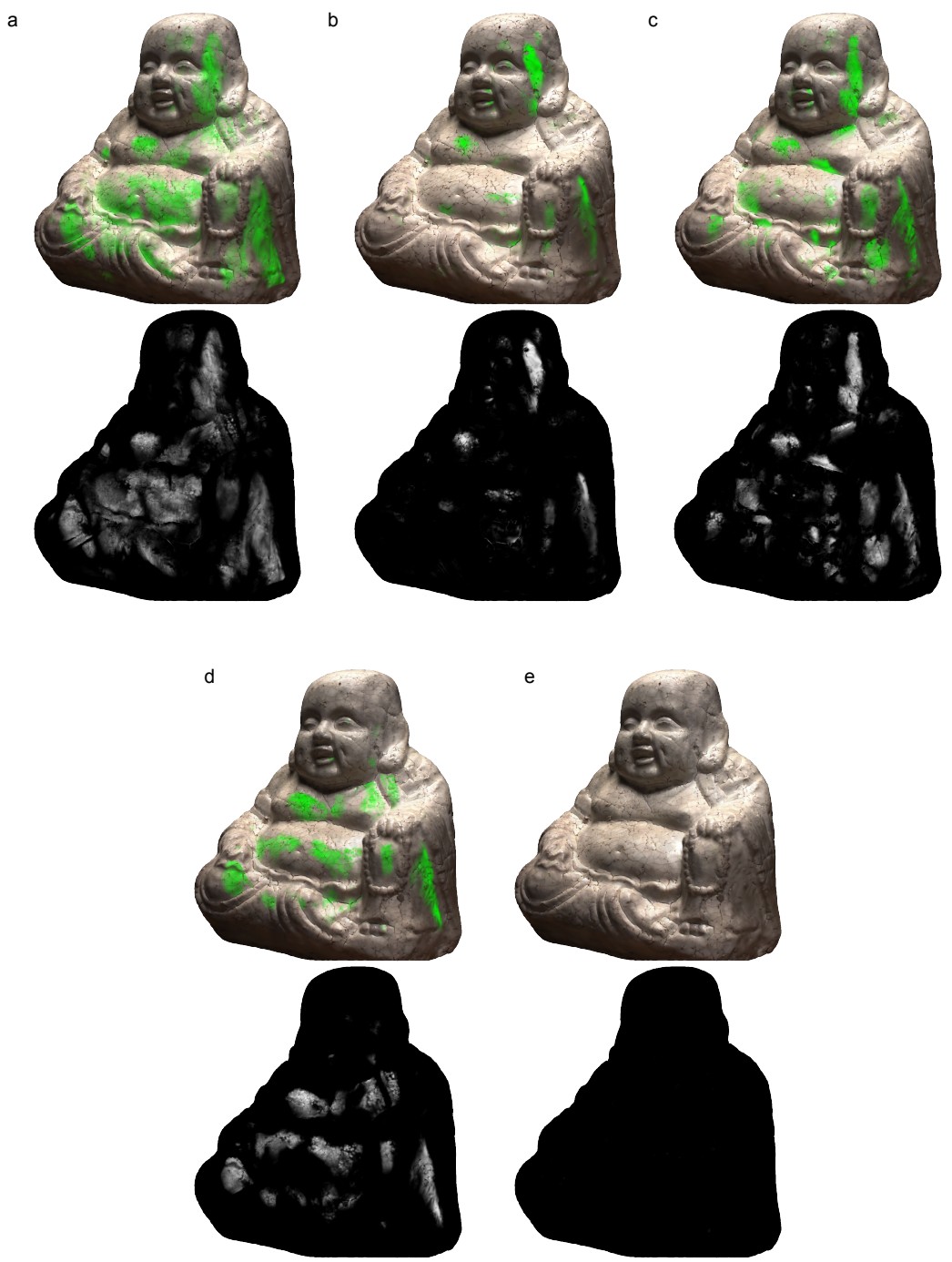

Figure S6: DTU renders with green infill revealing transparency (top) and corresponding transmittance maps (bottom) for (a) 3DGS, (b) GOF, (c) StopthePop, (d) Gsplat+$\alpha$ and (e) NGS.

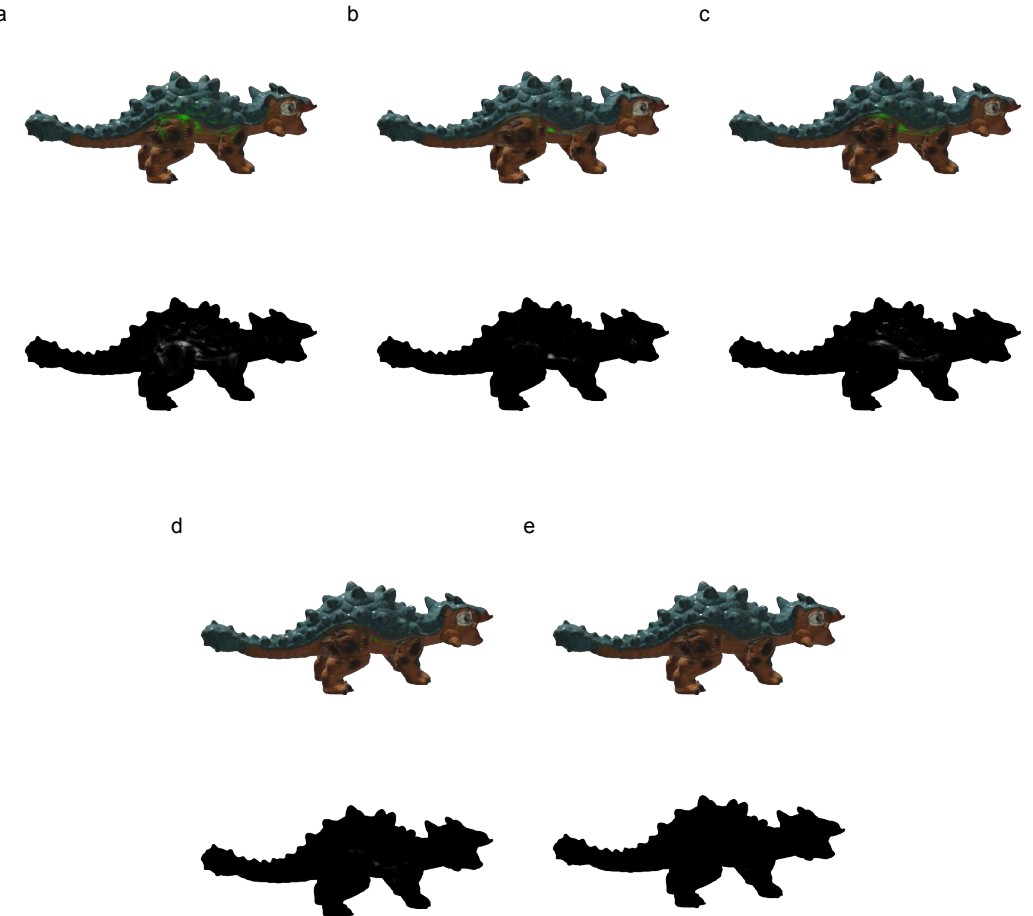

Figure S7: OmniObject3D renders with green infill revealing transparency (top) and corresponding transmittance maps (bottom) for (a) 3DGS, (b) GOF, (c) StopthePop, (d) Gsplat+$\alpha$ and (e) NGS.

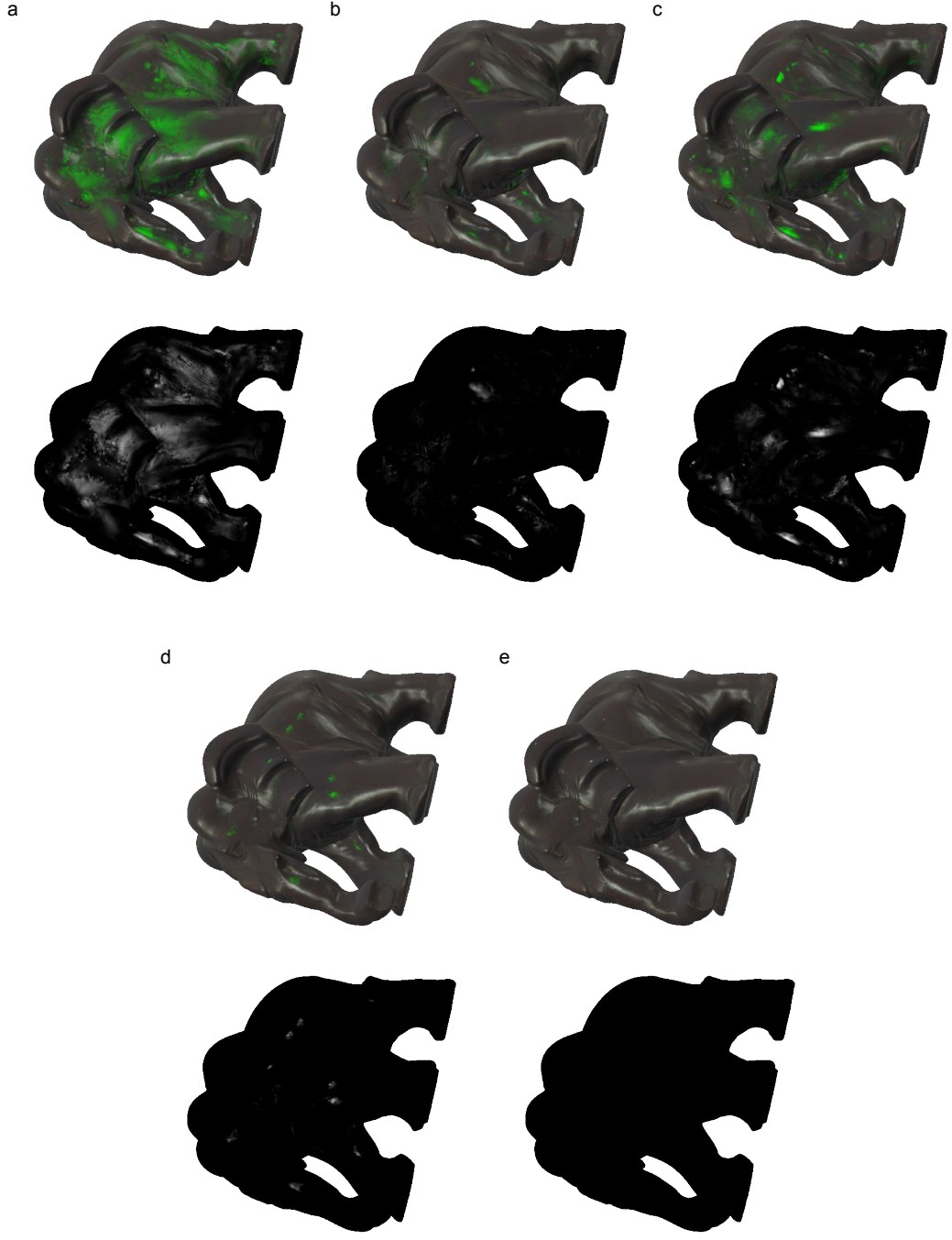

Figure S8: OmniObject3D renders with green infill revealing transparency (top) and corresponding transmittance maps (bottom) for (a) 3DGS, (b) GOF, (c) StopthePop, (d) Gsplat+$\alpha$ and (e) NGS.

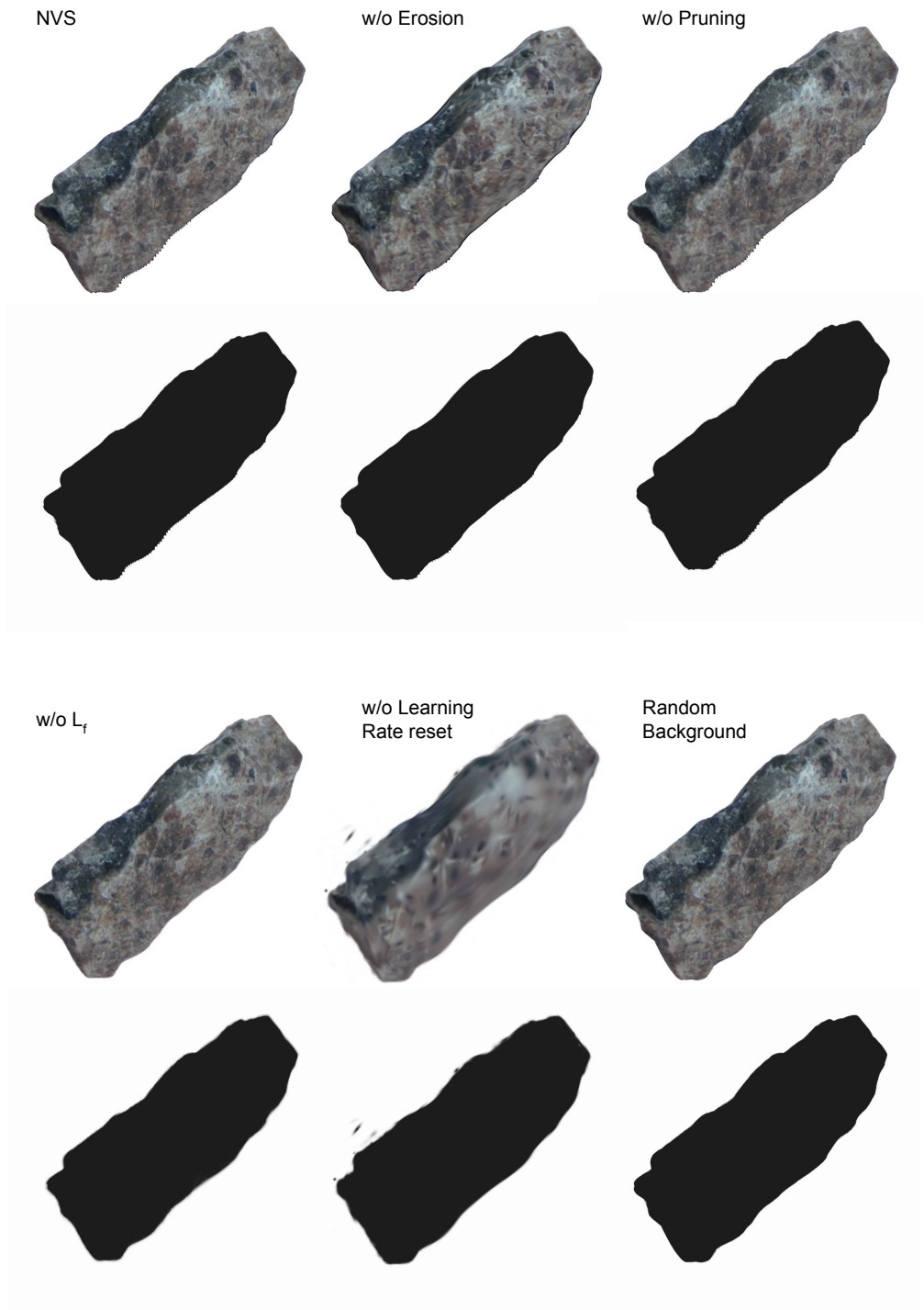

Figure S9: A sample from the *Stone Dataset* used in the ablation study. The reconstructions are filled with green noise. Reset the learning rate affects the quality of final rendering the most significantly. Stopping erosion also reduces the surface quality. Most of the results do not show visible false transparency, but the $SOS$ can be quantified and compared.

