# OpenReview forum: "Fix False Transparency by Noise Guided Splatting"
_NeurIPS.cc/2025/Conference — NeurIPS 2025 poster_

### Official Review · Reviewer_BBSm · 2025-06-29

**Clarity:** 1
**Significance:** 1
**Originality:** 2
**Rating:** 5
**Confidence:** 3

**Summary:**

This paper addresses the false transparency problem in 3D Gaussian Splatting by proposing a method called Noise Guided Splatting (NGS). The approach introduces (1) an alpha-consistency loss to suppress background opacity and (2) random noise Gaussians injected beneath the object’s surface during training to guide opacity optimization. Additionally, the paper presents a novel evaluation method for detecting transparency artifacts—often hard to perceive in static images—by infilling the interior with high-saturation colors for visualization and quantification.

**Questions:**

## 1. Is your false transparency problem the same as the floater artifact?
The false transparency issue described in your paper appears similar to the floater artifact, where surfaces are reconstructed at incorrect depths. Could you clarify whether your definition of false transparency is fundamentally different from floaters? If they are the same or closely related, please explain why your method should be considered novel in this context.

## 2. Could you re-express your method as an algorithm?
The current presentation is somewhat difficult to follow. A concise “Algorithm” box or pseudocode format would help readers clearly understand the overall training pipeline.
To confirm my understanding, here's my interpretation of your pipeline. Please correct or confirm:

```
For iteration in range(30000):
	If iteration == 6000:
		compute_convex_hull()
		prune_gaussians_not_farthest_in_depth()
	If iteration > 6000:
		assign_random_noise_color(RGBCMY)
	If iteration == 15k:
		_3DGS.require_grad = False
		_3DGS.opacity.require_grad = True
	Update_3DGS()
```
Is this an accurate reflection of your training process?

 ## 3. Is the pruning method potentially flawed?
From your description, it seems that pruning is done by retaining only the farthest Gaussian along each ray during depth testing. Wouldn’t this incorrectly remove valid front-surface points while keeping back-facing or internal points?

For example, given the scene setup as shown in [the picture](https://i.ibb.co/W47Sb3v0/image.png) that we try to reconstruct a scene of a ball. Let's say you try to prune the image of the ball. The ray hit 3 times. The floater, the front of the ball (close to the camera), and the back of the ball (far from the camera). But you only keep the farthest depth. The floater (shown as orange) will be removed as per your intention. But the surface on the front of the ball (shown as red) will be removed as well, and keep the back of the ball (shown as green), but in this case, you want to keep red.

![Image for question 3](https://i.ibb.co/W47Sb3v0/image.png)

Could you clarify or justify this pruning strategy?

## 4.  Is the Surface Opacity Score (SOS) perceptually aligned?

Could you provide visual comparisons illustrating what different SOS values (e.g., 0.95 vs 1.0) look like? In your ablation study, it seems that you favor a lower SOS (e.g., 0.95) over a perfect 1.0, which seems counterintuitive.

Additionally, since your method allows injecting infill into other models as shown in Figure 5, SOS can be applied to all baselines. Therefore, it would be helpful to include the SOS scores for all methods in Table 1 for a fairer comparison.

## 5.  Why limit noise colors to RGBCMY?
Why are noise Gaussians assigned only one of six fixed colors (Red, Green, Blue, Cyan, Magenta, Yellow) instead of using fully random RGB values? If this design choice is intentional, please explain the reason behind the design. It would also be helpful to include an experiment comparing RGBCMY vs. random RGB to demonstrate its impact, if any.
## 6. How is the 360° object case handled, given the limitations of alpha-consistency loss?
You mention that the alpha-consistency loss has limitations when applied to full 360° captures. However, both DTU and OmniObject3D are 360° datasets. Do you apply any adjustments or additional constraints in these cases? If not, how do you mitigate the failure of the loss in such scenarios?

**Ethical Concerns:**

["NO or VERY MINOR ethics concerns only"]

**Final Justification:**

The paper introduces three key contributions:

1. Identification and formalization of the false transparency problem in 3D Gaussian Splatting
2. An effective algorithm to address this issue
3. The Surface Opacity Score (SOS) metric for evaluating transparency artifacts.

These contributions are technically meaningful and justify acceptance.

Initially, the paper was difficult to follow due to fragmented presentation and missing context. However, the authors clearly show how they will restructure the method section and improve readability, which effectively addresses my earlier concerns.

**Limitations:**

yes

**Paper Formatting Concerns:**

No concerns. The paper follows the NeurIPS template, stays within the 9-page content limit, and appears properly formatted.

**Quality:**

1

**Strengths And Weaknesses:**

## Strength:
- **Quantitative evaluation of false transparency** The paper introduces a valuable method for visualizing and quantifying false transparency artifacts, which are difficult to detect in static renderings. Using high-saturation infill colors provides a clear and intuitive way to expose these issues during evaluation.

## Weakness:
- **Fragmented content structure:** The paper is difficult to follow due to its fragmented presentation. For example, Section 3.3 on Noise Guided Splatting (NGS) is divided into multiple loosely connected subsections. Some, such as the “Infill” subsection, focus on failed boundary estimation methods, which add little value. A more cohesive and step-by-step description of the full NGS pipeline would improve clarity.

- **Incomplete method description:** Key details are missing from the main method section. For instance, the learning rate reset is never introduced as part of the core pipeline, yet the ablation study later reveals it is essential for the method’s score improvement. Important training mechanisms should be clearly described up front in the method section.

- **Lack of distinction from floater artifacts:** The false transparency artifact appears closely related to the floater artifact. However, the paper does not clearly distinguish the two or cite recent related work on the floater (e.g., StableGS (arXiv 25.03), Optimize the Unseen (arXiv 24.12), Nerfbusters (ICCV 2023)). Without this clarification, the originality and scope of the contribution remain ambiguous.

- **Significance concerns**:  The method is limited to object-centric scenarios, which reduces its general applicability. Similar works on the floater artifact are compatible with general scenes, making them potentially more attractive to researchers seeking broader solutions.

---

> ### Author Rebuttal · Authors · 2025-07-31
>
> We appreciate the detailed feedback. We want to start by clarifying two key points that may have caused major confusion:
> - **Typo**: We mistakenly used an old abbreviation, "FTS," instead of our final metric name, Surface Opacity Score (SOS), in several tables. We apologize for this error and will correct it.
> - **Artifact Definition**: We didn't clearly define the "false transparency" symptom upfront, which may have led to confusion with "floater" artifacts.
>
> We address your specific points below.
>
> **Weakness 1: Fragmented content structure**
>
> The current presentation of our method in Section 3.3 could be more cohesive and step-by-step. We will include an overview in L138 with two major modifications: the introduction of noise Gaussians within the object’s volume in a coarse to fine approach and depth aware noise gaussian pruning and erosion in iteration t1 allowing enough time to establish basic surface structure and appearance, and the addition of alpha consistency loss. The infill step is indeed important for noise injections.
>
>
> **Weakness 2: Incomplete method description**
>
> Good point. The learning rate (LR) reset is an important detail. While we included it in the implementation details and ablation studies, we agree it belongs in the main method description. We'll add it to Section 3 in the revision for better clarity.
>
> **Weakness 3 / Q1: Distinction from floater artifacts**
>
> We want to clarify that false transparency is fundamentally different from floaters, even though both can lead to visual inconsistencies in interactive viewing and are not well-captured by standard static rendering metrics.
> - Floaters involve sparse points reconstructed at incorrect depths above the surface.
> - False transparency occurs when the opaque surface is reconstructed at the **correct depth** with **wrong opacity**
> - Floaters can show up in static novel view renderings, not the case for false transparency
> - NGS does not solve floater problem
> - Infill and our datasets do not reveal the floater artifact
> - SOS does not measure the floater artifact
> - NGS should be compatible with methods that address floater problem
>
> This leads to different symptoms. The false transparency artifact is only apparent during viewport transition, where background and internal details seem to move out-of-sync with the surface, creating an illusion of translucency. Floaters, on the other hand, are often visible as detached blobs in static novel-view renderings. Our method (NGS), our evaluation metric (SOS), and our dataset are all designed to address opacity and do not solve or measure the floater problem. We'll add a section to the related works to explicitly discuss this distinction.
>
> **Weakness 4: Significance concerns**
>
> Regarding your concerns:
> - **Limitation**: It's the artifact itself, not our method, that is most relevant to solid, object-centric scenarios. Unlike floaters, which can appear anywhere, false transparency is an issue of objects not appearing solid when viewed from all sides.
> - **Significance**: The significance of this paper isn't just the solution. Our primary contribution is being the first to **identify, define, solve, and provide tools to measure** the previously unreported "false transparency" artifact. By raising awareness and contributing new evaluation methods (the infill technique and SOS metric), we believe our work will enable and facilitate future research in this area.
>
>
> **Q2: Could you re-express your method as an algorithm?**
>
> The key steps missing from the initial algorithm were the noise injection and the distinction between surface and noise Gaussians. Here is a more accurate representation:
> ```
> for iteration in range(30000):
>     if iteration == 6000:
>         # Initialize and Prune Noise Gaussians
>         noise_gaussians = compute_convex_hull_for_surface_saussians_and_voxelize()
>         depth_prune_and_binary_erode(noise_gaussians)
>         freeze_parameters(surface_gaussians)
>         for iteration i from 1 to 1000:
>             refine_prune_based_on_opacity(noise_gaussians)
>             reset_learning_rate(surface_gaussians.means)
>             freeze_parameters(noise_gaussians)
>             unfreeze_parameters(surface_gaussians)
>     if iteration > 6000:
> 	assign_random_RGBCMY_color(noise)
> 	render_image_with_both_gaussian_sets(surface_gaussians + noise_gaussians)
> 	compute_photometric_loss()
> 	compute_alpha_consistency_loss(surface_gaussians, seg_mask)
> 	update_parameters(surface_gaussians)
> ```
>
> **Q3: Is the pruning method potentially flawed?**
>
> This is an excellent question that stems from the confusion we created. We apologize for the ambiguity. The pruning does not touch the original surface Gaussians. It is applied only to the newly injected noise Gaussians to ensure they are correctly positioned inside the object and don't interfere with the surface. The process is:
> 1. We inject a cloud of noise Gaussians inside the object's approximate volume.
> 2. We then prune any of these noise Gaussians that were accidentally initialized in front of or too close to the surface Gaussians.
> 3. The remaining noise Gaussians form an internal "infill" that forces the actual surface to become opaque.
>
> Using the provided figure as an example:
> - The figure misses the injected noise gaussians.
> - The noise gaussians are injected inside the ball approximately between the red X and green X.
> - The pruning is performed on the injected noise gaussians.
> - Noise gaussians above or near the object surface will be pruned.
> - Noise gaussians inside the object cavity are not pruned and become the infill that indicates the space.
> - The "floater" artifact (orange X in the figure) is irrelevant to our pruning step, as it's part of the surface model, which we don't prune.
>
>
> **Q4: Is the Surface Opacity Score (SOS) perceptually aligned? And why does ablation favor lower SOS?**
>
> Yes, SOS is designed to be perceptually aligned. A score of 1.0 is fully opaque, and a score near 0.0 is fully transparent. We find that renders with an SOS above 0.5 generally have no visible false transparency. A figure will be provided to demo perceptual linearity.
>
> Regarding the ablation in Table 2, the "w/o Erosion" setting shows a trade-off. While its SOS is a marginal 1.0 vs. our 0.968, this comes at a significant cost to standard rendering metrics (PSNR/SSIM). Our method achieves the best balance between high opacity and overall render quality.
>
> And to reiterate, we deeply apologize for the "FTS" vs. "SOS" typo in the tables. It will be corrected everywhere.
>
>
> **Q5: Why limit noise colors to RGBCMY?**
>
> We chose RGBCMY because these six primary and secondary colors offer maximum contrast against most natural surface colors in RGB and HSV space. This high contrast creates a strong, unambiguous error signal if any noise "leaks" through a semi-transparent surface, forcing the model to quickly optimize the surface opacity to hide it.
>
> As per your suggestion, we ran an experiment with random RGB noise. The results confirm our initial design choice, as RGBCMY provides better overall performance. We'll add this to our ablation study.
>
>
> | Method               | PSNR    | PSNR\*  | SSIM   | SSIM\* | LPIPS  | LPIPS\* | SOS    |
> | :------------------- | :------ | :------ | :----- | :----- | :----- | :------ | :----- |
> | default              | **32\.201** | **32\.201** | **0\.934** | **0\.934** | **0\.145** | **0\.145**  | **0\.969** |
> | ...          | ... | ... | ... | ... |... | ...  | ... |
> | random RGB noise        | 31\.442 | 31\.436 | 0\.926 | 0\.926 | 0\.154 | 0\.154  | 0\.962 |
>
>
> **Q6: How is the 360° object case handled, given the limitations of alpha-consistency loss?**
>
> You've correctly identified a key synergy in our method. While standard alpha-consistency loss can struggle with 360° captures, **NGS directly solves its primary limitation**.
>
> The injected noise acts as an internal **occlusion barrier**, physically separating the front and back surfaces of the object. This provides the strong bias for opacity on the front-most surface that the alpha-consistency loss alone lacks in a 360° setting. This is why NGS performs well on 360° datasets like DTU and OmniObject3D, as shown by the strong SOS scores in our results.

---

> ### Comment · Reviewer_BBSm · 2025-08-02
> **Clarified key concerns; novelty remains, but clarity still a concern**
>
> Thank you for the detailed clarifications. I now understand how your work differs from floater artifacts, and I agree with your responses to all of my questions.
>
> The paper retains its novelty, particularly in:
> - The proposed algorithm to mitigate false transparency (Q2),
> - The Surface Opacity Score metric
>
> After your explanation, I also no longer consider the concern raised in Q3 to be a technical fault.
>
> The proposed revisions—rewriting Section 3.3 in a clearer step-by-step format and adding an ablation on random RGB noise—are promising improvements.
>
> I encourage you to explicitly state the distinction from floater artifacts in the main paper or supplementary material and to better frame the false transparency problem in the introduction.
>
> While I no longer see a strong reason to reject, I remain cautious about whether the promised revisions will sufficiently improve clarity.
>
> Current rating: 4 – Borderline Accept
> Note: I remain open to adjusting my score based on further discussion.

---

> ### Author Response · Authors · 2025-08-04
> **Proposed edits to the original manuscript**
>
> - **Improve our abstract by better describing the symptom of the artifact and clarify false transparency is a new type of artifact (in \<\>):**
>
> \<Opaque objects reconstructed by 3D Gaussian Splatting (3DGS) often exhibit a falsely transparent surface. Camera movement in interactive viewing reveals constantly changing internal and background patterns beneath the surface.\> This issue stems from the ill-posed optimization of 3DGS…
>
> … false transparency has not been explicitly identified. \<In this work, we are the first to identify, characterize, and develop solutions for this "false transparency" artifact, an underreported artifact in 3DGS. Our strategy, Noise Guided Splatting (NGS), encourages surface Gaussians to adopt higher opacity by injecting opaque noise Gaussians in the object volume during training and minimally modifying the existing splatting process.\> To quantitatively evaluate the false transparency …
>
> - **Before the 2nd paragraph of the introduction, clarify the symptom of false transparency before discussing causes:**
>
> False transparency causes opaque surfaces to incorrectly appear semi-transparent, exclusively observable during interactive viewing and undetectable in individual frames and standard IQA metrics. The artifact manifests as a parallax-induced transparency effect: during camera movement, objects exhibit a disturbing "see-through" quality where internal and background Gaussian structures become visible through surfaces that should be opaque. These internal structures move out of alignment with the surface during rotation, creating an illusion reminiscent of frosted glass. The effect is most severe in objects with homogeneous surfaces (e.g., stones, food items, ceramics, plastics, etc.), where limited texture variation provides insufficient constraints for correct opacity optimization.
>
> This false transparency artifact occurs because 3DGS is supervised primarily by a 2D photometric loss between rendered and ground-truth images…
>
> - **Modify Sec 2 - Related works - View inconsistency:**
>
> **NVS artifacts.** There are several well recognized NVS artifacts that should not be confused with false transparency artifacts . A common one is floater artifacts, which manifest as sparse features reconstructed at incorrect depths above the surface \[StableGS, Optimize the Unseen\]. These artifacts do not appear in training views but become obvious in novel views. Floaters have been successfully mitigated using depth consistency constraints \[StableGS, RaDe-GS\] and specialized priors \[Unseen, Nerfbusters, HD-GS\]. Another category is view-inconsistency artifacts, which cause surfaces to exhibit unnatural changes during viewpoint transitions. A well-known example is the ‘popping artifact’ \[6\], caused by sorting discontinuities between adjacent views. Hierarchical sorting \[6\], order-independent transparency \[2\], anti-aliasing filtering \[4\], and hybrid transparency \[5\] have been introduced to address these problems. Finally, 3DGS also suffers from poor reconstruction of certain details, which has been addressed using specialized loss functions \[E-Rank, Neurips 2024\] and diffusion-based post-processing enhancements \[difix3d+, CVPR 2025\].
>
> - **The introductory paragraph to Sec 3.3 will present a more clear overall workflow outline:**
>
> NGS addresses false transparency by employing an alpha-consistency loss strategically placing noise Gaussians within the object's volume to obstruct direct lines of sight between front and back surfaces, thereby forcing the optimization to prioritize an opaque foreground. The pipeline begins by **initializing** a set of noise Gaussians within a coarse voxelized convex hull of the object. The color of the noise gaussians are randomized in each iteration to prevent overfitting. These noise Gaussians are then **pruned**, ensuring only noise Gaussians inside the object remain. This initialization and pruning sequence is repeated in a **multi-scale** manner across increasing voxel resolutions to accurately fill complex geometries while remaining memory efficient. Afterwards, we conduct a brief **fine tuning** phase where the surface Gaussians are frozen, and the noise opacities are trained and pruned. Finally, the noise Gaussians are frozen, the surface Gaussians are unfrozen and training proceeds normally with a reset learning rate.
>
> - **Remove the confusing infill subsection, and the remaining sections should correspond to the key steps**
>
> - **Discuss the choice of RGBCMY color in the Fine tuning section by adding:**
> RGBCMY is chosen because these six primary and secondary colors offer maximum contrast against most natural surface colors in RGB and HSV space.
>
> - **Replace the previous version of Table 1 with a corrected version with better clarity (see rebuttal to review 6z1z), as well as tables in appendix**
>
> - **Add random RGB noise to the ablation study Table 2**
>
> Happy to add more clarification if the reviewer has additional recommendations.

---

> > ### Comment · Reviewer_BBSm · 2025-08-05
> > **Paper is now clearer; I have no remaining concerns**
> >
> > Thank you for showing how the paper will be revised. I believe the updated significantly improves clarity
> >
> > One minor suggestion: the term IQA (Image Quality Assessment?) might not be familiar to all readers. Using the full term at first mention would enhance readability.
> >
> > Overall, I have no remaining concerns.
> >
> > The paper introduces three key contributions:
> >
> > 1. Identification of the false transparency problem
> > 2. An effective algorithm to address it
> > 3. The Surface Opacity Score (SOS) metric for evaluation.
> >
> > These represent meaningful technical contributions that justify acceptance.
> >
> > **Current rating:** 5 - accept

---

### Official Review · Reviewer_N48H · 2025-06-30

**Clarity:** 3
**Significance:** 2
**Originality:** 3
**Rating:** 3
**Confidence:** 3

**Summary:**

The paper spotlights a “false-transparency” artefact in 3DGS that causes opaque surfaces to appear semi-transparent when front- and back-layer Gaussians blend. It proposes Noise-Guided Splatting (NGS), which injects interior noise Gaussians so the optimiser is forced to sharpen surface opacity. A new Surface Opacity Score (SOS) and the stone dataset are introduced to quantify improvements. NGS roughly doubles SOS while keeping PSNR/SSIM stable.

**Questions:**

1. Please provide quantitative evidence that eliminating false transparency truly improves 3D reconstruction.
2. The current draft convincingly shows that Noise-Guided Splatting raises opacity, but it remains unclear whether this benefit translates into a leaner and more surface-aligned Gaussian representation. I strongly encourage you to include an explicit Gaussian-count vs. quality analysis. If NGS achieves the same reconstruction quality with fewer Gaussians (excluding noisy Gaussians), this would demonstrate a tangible efficiency upside.
3. What is the metric "FTS" in Table 1? It's not mentioned in the paper.

**Ethical Concerns:**

["NO or VERY MINOR ethics concerns only"]

**Final Justification:**

Regarding the core solution, I agree with the authors' explanation and the method is proved to be effective. However, I believe that providing a stronger theoretical foundation would significantly enhance the method’s credibility and generalizability.

At present, the solution appears highly dependent on some assumptions such as full opacity, the availability of a good initial point cloud, and the exclusion of thin structures, as replied by the author, which may limit its applicability in broader, more unconstrained scenarios.

Moreover, while the method addresses a genuine artifact in 3DGS and proposes novel evaluation metrics SOS and infill drop, the practical impact remains unclear. If the method does not consistently improve reconstruction quality or NVS performance, its contribution to the community may be limited. A more compelling case is needed to demonstrate why solving false transparency is essential, especially in real-world applications, and how this solution integrates with or benefits downstream tasks in 3D vision.

In its current form, I remain unconvinced about the broader value of the method beyond the specific diagnostic use case. I will keep my rating to 3, borderline reject.

**Limitations:**

yes

**Paper Formatting Concerns:**

No.

**Quality:**

3

**Strengths And Weaknesses:**

### Strengths
1. The paper tackles a genuine and under-explored artifact, spurious transparency, in 3DGS, giving the topic practical relevance.
2. The proposed Noise-Guided Splatting (NGS) maintains parity with baselines on standard image metrics while markedly outperforming them on a dedicated transparency score.

### Weaknesses
1. The solution is essentially an empirical “noise fill-in” heuristic; the manuscript offers no theoretical rationale or analysis to explain why it should eliminate false transparency or under what conditions it might fail.
2. NGS yields only parity on NVS quality, yet incurs ~50% extra memory and additional training time. The authors argue that better opacity should benefit novel-view synthesis and reconstruction, but provide no downstream reconstruction metrics to substantiate this claim.
3. The newly introduced “Stone” dataset consists of niche, single-material objects, limiting its representativeness; thus, the generality of the transparency findings remains unclear.

---

> ### Author Rebuttal · Authors · 2025-07-31
>
> We are grateful for your thorough review and insightful comments. We address your specific points below:
>
> **Q3: What is the metric "FTS" in Table 1?** *(We want to address this first, as it is a critical point. )*
>
> **"FTS" is a typo and should be Surface Opacity Score (SOS)**. We initially used the term "False Transparency Score" (FTS) but changed it to SOS for better perceptual alignment, where a higher score correctly indicates a more opaque surface. We sincerely apologize for failing to update this term in the table and for the confusion it caused.
>
> **Weakness 1.1: The solution is an empirical "noise fill-in" heuristic lacking theoretical rationale/analysis**
>
> We agree the theoretical rationale was not sufficiently clear. The root cause of false transparency is an ill-posed optimization problem (Sec 3.1, Fig 2). Because 3DGS training relies on 2D photometric loss, two scenarios are ambiguous:
> 1. An opaque surface
> 2. A semi-transparent surface blended with background or internal Gaussians
>
> Both can produce identical static images, allowing the optimizer to erroneously favor the second case. Our method, NGS, resolves this ambiguity by injecting noise Gaussians inside the object. This noise acts as an **occlusion barrier**, breaking the line of sight from the camera to the background. The photometric loss is now forced to make the surface opaque to hide the random colors of the internal noise. These points are briefly discussed in  Section 3.3 (L135), but we believe this section requires more substantial rationale.
>
> While NGS is a heuristic, its empirical nature is a strength, making it an adaptable, **"plug-and-play"** solution that minimally alters the core rendering pipeline. Furthermore, by defining the object's interior, NGS enables our proposed SOS and infill evaluation methods, which are valuable tools for characterizing this artifact in any method.
>
> **Weakness 1.2: under what conditions it might fail?**
>
> A primary failure mode is the integration of noise into the object's surface, which would degrade reconstruction quality. We did not state this explicitly, but several core components of NGS are designed specifically to prevent this:
> - Depth-aware pruning
> - Erosion of the occupancy volume
> - Trainable-only noise opacity
> - Randomized noise color
>
> We will clarify this in the revision. Other limitations of NGS, discussed in L299, include:
> - assumes full opacity and may fail on inherently translucent objects
> - requires a reasonable initial point cloud to define the convex hull for noise injection
> - less effective for very thin structures where injecting noise is difficult (though false transparency is also less common in such cases).
>
> **Weakness 2.1: NGS yields only parity on NVS quality but incurs ~50% extra memory and additional training time**
>
> We acknowledge this concern. The ~50% overhead reported was for our controlled experiments and is not an optimized lower bound. This cost is highly tunable. As noted in L211, memory and time can be substantially reduced by lowering the noise density or resolution. Removing Spherical Harmonics (SH) parameters from noise Gaussians is a strategy we did not mention, which alone would reduce their memory footprint by ~80%. These were not implement as we only uses 4GB VRAM on average.
>
> **Weakness 2.2: Lack of downstream reconstruction metrics to substantiate claims of improved 3D reconstruction**
>
> We wish to reiterate that the false transparency artifact is **undetectable with static renderings and standard NVS metrics**. The quality improvement is in the **continuous view coherence** of the 3D reconstruction, which is only apparent during interactive viewing.
>
> False transparency manifests as a distracting visual artifact where background and internal structures move out-of-sync with the surface, making it appear translucent during camera motion. Fixing this ensures a stable, solid surface, which is a crucial aspect of 3D reconstruction quality not captured by static 2D metrics. A substantial contribution of our work is providing the first reconstruction metrics to characterize and measure this previously overlooked dynamic artifact.
>
> **Weakness 3: "Stone" dataset's niche, single-material nature limits representativeness, thus generality of transparency findings unclear**
>
> We acknowledge the specific nature of the Stone dataset. It was collected due to its challenging properties (low texture, repeating patterns) that make false transparency artifacts more prominent, establishing it as an ideal diagnostic tool.
>
> However, our findings are not limited to this dataset. We demonstrate that false transparency is a general problem by providing analysis and infill add-ons for the diverse DTU and OmniObject3D datasets (see Appendix Fig. S2-S5). To further prove the generality of our findings, we have since scanned 100+ common objects (fruits, vegetables, toys, office supplies, kitchen supplies, sport supplies, etc.) and observed the same artifact. We will release this expanded dataset to support future research.
>
> **Q1: Please provide quantitative evidence that eliminating false transparency truly improves 3D reconstruction**
>
> As **standard NVS metrics are insensitive to this dynamic artifact**, we proposed two new quantitative evaluation strategies:
> - **Measure with/without the infill**: As shown in Table 1, a large drop in PSNR/SSIM between a normal render (e.g., PSNR) and a render with our noise infill visible (PSNR*) indicates significant transparency, as the "leaked" infill pattern degrades image quality.
> - **Surface Opacity Score (SOS)**: This is a direct, quantitative measure of surface opacity designed to assess false transparency without relying on standard metrics.
>
> We invite the reviewer to see the revised Table 1 with the latest experiments results and corrected terms in our rebuttal to reviewer 6z1z.
>
> **Q2.1: Does this benefit translate into a more surface-aligned Gaussian representation?**
>
> NGS does not explicitly enforce surface-aligned Gaussians. Its goal is to solve the opacity ambiguity. However, it is fully compatible with orthogonal methods that do promote surface alignment (e.g., RaDe-GS, 2DGS).
>
> **Q2.2: Leaner Gaussian? explicit Gaussian-count vs. quality analysis**
>
> This is an excellent suggestion. While reducing the Gaussian count was not a primary goal, we have conducted this analysis as suggested. We will include a Gaussian-count vs. quality comparison in the revised manuscript.
>
> |Data|Method|PSNR|SSIM|LPIPS|SOS|Gaussian Count|
> |:---|:---|:---|:---|:---|:---|:---|
> |DTU|NGS(gsplat+$L_{\alpha}$+noise)|25.428|0.881|0.192|**0.749**|**712325**|
> ||gsplat+$L_{\alpha}$|**25.435**|**0.884**|**0.183**|0.598|1046790|
> |Stones|NGS(gsplat+$L_{\alpha}$+noise)|**34.148**|**0.951**|**0.053**|**0.922**|572770|
> ||gsplat+$L_{\alpha}$|33.832|0.948|0.062|0.891|**467744**|
> |OmniObject3D|NGS(gsplat+$L_{\alpha}$+noise)|**33.619**|0.972|**0.060**|**0.736**|**219470**|
> ||gsplat+$L_{\alpha}$|33.575|**0.973**|**0.060**|0.642|278457|
>
> Interestingly, inclusion of noise in the training indeed led to a leaner gaussian representation with an average decrease of 30% on DTU and OmniObject3D.  The gaussian count is higher for the stone dataset, but the reconstruction metrics are also substantially higher.

---

> > ### Comment · Reviewer_N48H · 2025-08-05
> >
> > Thank you for the detailed rebuttal. I appreciate the authors’ efforts.
> >
> > Regarding the core solution, I agree with the authors' explanation and the method is proven to be effective. However, I believe that providing a stronger theoretical foundation would significantly enhance the method’s credibility and generalizability.
> >
> > At present, the solution appears highly dependent on some assumptions such as full opacity, the availability of a good initial point cloud, and the exclusion of thin structures, as replied by the author, which may limit its applicability in broader, more unconstrained scenarios.
> >
> > Moreover, while the method addresses a genuine artifact in 3DGS and proposes novel evaluation metrics SOS and infill drop, the practical impact remains unclear. If the method does not consistently improve reconstruction quality or NVS performance, its contribution to the community may be limited. A more compelling case is needed to demonstrate why solving false transparency is essential, especially in real-world applications, and how this solution integrates with or benefits downstream tasks in 3D vision.
> >
> > In its current form, I remain unconvinced about the broader value of the method beyond the specific diagnostic use case.

---

> ### Author Response · Authors · 2025-08-06
>
> We appreciate that the reviewer acknowledges the effectiveness of the method. We want to clarify some additional misunderstandings:
> - **Theoretical foundation:** NGS is  developed based on a strong theoretical foundation on the root cause of the problem: the ill-posed optimization process fails to distinguish the foreground subface, intermediate space, and the background surface in discrete views, which causes an alpha ambiguity (**Section 3.1, Eq. 1-3, Fig. 2**). Breaking line-of-sight is a straightforward and effective strategy to regularize the ill-posed problem, and NGS's effectiveness also supports this theory. We are open to specific suggestions to strengthen this theoretical foundation.
> - **Applicability in broader, more unconstrained scenarios:** We want to clarify in most cases it is not the method's applicability is limited. False transparency itself does not apply to certain scenarios.
>     1. **Opaque object:** The fact that this artifact is "False Transparency" requires the object to be opaque. Accurately speaking, **opaque object is a classification rather than an assumption.** It should not be implemented in translucent objects. A substantial portion of natural/daily objects are opaque, which applies to almost **100% of the data** in popular datasets such as DTU and OmniObject3D. It also applies to a vast majority of cases in larger scale 3D scan datasets such as Common/Uncommon Objects in 3D and Project Aria. On the project page of Uncommon Objects in 3D, **>90%** interactive data meet this criterion and **>50%** show some level of false transparency.
>     2. **Good initial point cloud:** We would like to clarify that it is not our method alone that requires a good initial point cloud. A good initial point could be a universal requirement of most current splatting-based methods. Poor initial point cloud would cause poor reconstruction of the underlying 3DGS method itself, not just NGS.
>     3.  **Exclusion of thin structures:** To clarify, our method does not exclude objects containing thin structures. The method elegantly skips thin structures and applies to the rest of the object. In case of thin structures, the foreground surface and background surface are close to each other, so splatting-based methods tend to blend them together as a single layer. Thus, **false transparency is uncommon and less obvious in such cases.** Our pruning and erosion steps are designed to automatically remove noise Gaussians that would interfere with these thin parts of an object but not the rest of the large object, ensuring the method's effectiveness is concentrated where it is most needed.
>
> - **Practical impact:**
>    1. **Improve reconstruction quality or NVS performance:** We would like to clarify this point again. In **Weakness 2**, the reviewer argued we provided no downstream reconstruction metrics to substantiate this claim. To re-iterate, traditional reconstruction quality or NVS performance benchmark fails to detect and measure false transparency. They do not reflect all aspect of reconstruction quality. NGS **consistently improves reconstruction quality and NVS performance by eliminating the artifact**. Fixing false transparency does not directly improve the traditional reconstruction quality or NVS performance scores, **because the theoretical foundation indicates false transparency is a valid optimal solution to the ill-pose optimization process**. To demonstrate our effectiveness, we introduced new, infill-augmented metrics (PSNR*, SSIM*, LPIPS*) and the Surface Opacity Score (SOS), which are specifically designed to quantify this artifact and on which our method shows significant improvement.
>     2. **Applications:** Object-centric scans are widely used as 3D assets in gaming, movies, advertisement, and other VFX applications. They are the most common type of 3D scan assets. Although mesh-based 3D models are still dominating, splatting-based models are gaining significant traction more recently due to its ability to directly encode light. Many production tools start to support 3DGS assets. For these applications, consistent dynamic visual quality is extremely important and can directly affect user experience. Artifacts like false transparency will make the assets appear unreal and degrade the quality of the production. 3DGS assets are also promising for AR/VR applications. False transparency artifact becomes distracting and prominent in a head-mounted display. We recognize these applications might not be clear to readers from different domains, and we will clarify the use case in the revision.
>     3. **Diagnostic use case:** We want to argue that **identifying the problem, raising the awareness and enabling detection/quantification with new tool alone should be considered important practical impacts.** This is also acknowledged by some other reviewers. We hope our findings can support future research and we are open to suggestions to highlight this aspect of work.

---

### Official Review · Reviewer_6z1z · 2025-07-02

**Clarity:** 2
**Significance:** 2
**Originality:** 2
**Rating:** 3
**Confidence:** 4

**Summary:**

The paper addresses the problem of false transparency artifacts in 3D Gaussian Splatting, where opaque surfaces appear incorrectly semi-transparent due to ill-posed alpha blending. A method called Noise Guided Splatting (NGS) is proposed to solve the problem. NGS injects internal noise Gaussians during training to enforce correct surface opacity. The paper further introduces a transmittance-based metric and a new diagnostic dataset. Experiments on multiple object-centric datasets demonstrate that NGS can effectively reduce false transparency without degrading rendering quality.

**Questions:**

(1). I am curious about how much the reported false transparency issue affects rendering quality, especially in interactive or dynamic scenarios. The problem of false transparency should be defined more precisely, with clearer examples illustrating how it degrades rendering quality in static and interactive scenes.

(2). Additional qualitative visualizations on standard open datasets (e.g., DTU and OmniObject3D) would better support the claims and show the method’s effectiveness.

(3). The authors should clarify whether the injected noise infill would introduce new artifacts or unintended occlusion, and how the method avoids damaging fine surface details.

(4). The generalization of the proposed method to more complex scenes should be discussed, especially regarding how the noise infill is initialized and managed in such cases. Current result visualization only focuses on single object-centric scenarios.

(5). There are some typos and figure errors that need to be corrected.

**Ethical Concerns:**

["NO or VERY MINOR ethics concerns only"]

**Final Justification:**

I have read the authors’ rebuttal as well as the other reviewers’ comments.  The rebuttal clarifies some terms and descriptions, which reveals that the paper indeed contains ambiguities and errors. In the rebuttal, the authors emphasize that a central contribution of the work is the identification and analysis of "false transparency", which occurs primarily during interactive viewing. Then a straightforward question arises: how important is this issue, especially in the context of 3D-GS? This significance is neither clearly explained nor well illustrated in the paper. Since the evaluation should mainly be based on the submitted version, I maintain my original rating. However, given that the authors could improve their paper by incorporating the comments or feedback from the reviewers, I would also consider a Borderline accept reasonable.

**Limitations:**

The paper has discussed the main limitations of the proposed method.

**Paper Formatting Concerns:**

NIL

**Quality:**

3

**Strengths And Weaknesses:**

Strengths:

The paper presents a practical and easy-to-implement approach to mitigate false transparency artifacts in 3D Gaussian Splatting. The method can be integrated seamlessly into existing pipelines.  It is systematically evaluated with the help of a new dataset. The introduction of a quantitative metric and dataset also provides a useful tool for analyzing the artifact in future research.


Weaknesses:

The main issue with the paper is that the authors do not clearly identify or exemplify the false transparency problem before introducing their solution. Although they point out that the problem is more apparent in dynamic renderings, the paper does not provide clear dynamic or temporal results to support the claim. The effectiveness of the proposed method on standard datasets is not convincingly demonstrated and lacks comparison with strong baselines.

---

> ### Author Rebuttal · Authors · 2025-07-31
>
> We appreciate the reviewer's recognition of our method's practicality, ease of implementation, and the utility of our proposed metric and dataset. We wish to re-emphasize that one of our primary goals is to introduce the "false transparency" problem, a previously unreported issue that is difficult to detect with conventional metrics. Our work provides the first identification of this problem, an effective solution, and practical tools (metrics, dataset) to evaluate it and facilitate future research.
>
> **Weakness 1.1: missing clear dynamic or temporal results to support the claim**
>
> This point stems from a linguistic misunderstanding. In lines 37-38, we intended to distinguish between:
> - **Static images**: Rendered 2D images from the model
> - **Dynamic renderings**: Interactively viewing the static 3D model in a viewer (e.g., Viser).
>
> Our terminology failed to distinguish this from rendering a 4D dynamic scene. To correct this ambiguity, we will revise the phrase “dynamic renderings” to the more precise “interactive viewing of 3DGS models” throughout the manuscript.
>
>
> **Weakness 1.2: the authors do not clearly identify or exemplify the false transparency problem**
>
> We agree that we did not sufficiently exemplify this novel problem upfront, which is critical for a previously unrecognized issue. We will revise the manuscript to provide a clearer, more detailed description of the problem from the start:
> - **Abstract**: We will clarify that "false transparency" is a novel problem and that this paper provides its identification, solution, and evaluation framework.
> - **Introduction**: We will describe the symptoms more clearly. False transparency manifests during interactive rotation, where background and internal structures appear to "show through" the object's surface. This illusion occurs because sparse internal Gaussians, which are blended and hidden in static views, move out of sync with the surface during rotation. This parallax effect creates a distracting, translucent appearance, especially when fine background textures misalign with the surface.
>
> **Weakness 2: The effectiveness of the proposed method on standard datasets is not convincingly demonstrated and lacks comparison with strong baselines.**
>
> After revising the comments, we have realized that Table 1 did not include the latest version of experiments and had mislabeled data, noticeably including the old term FTS instead of Surface Opacity Score (SOS), which was our previous name for the new metric. We have updated the table and provide a revised version with corrected SOS label.
>
> |Data|Method|PSNR|PSNR*|SSIM|SSIM*|LPIPS|LPIPS*|SOS|
> |:---|:---|:---|:---|:---|:---|:---|:---|:---|
> |DTU|3DGS|25.575|22.967|**0.891**|0.874|**0.180**|0.250|0.147|
> |DTU|GOF|**25.648**|21.109|0.88|0.816|0.209|0.273|0.179|
> |DTU|StopThePop|22.817|18.885|0.852|0.78|0.213|0.315|0.135|
> |DTU|gsplat + $L_{\alpha}$|25.435|25.263|0.884|**0.883**|0.183|**0.186**|0.598|
> |DTU|NGS (gsplat + $L_{\alpha}$ + noise)|25.428|**25.427**|0.881|0.881|0.192|0.192|**0.749**|
> ||
> |Stone|3DGS|**34.61**|27.551|0.949|0.909|0.055|0.222|0.14|
> |Stone|GOF|31.469|21.998|0.893|0.78|0.186|0.324|0.126|
> |Stone|StopThePop|32.457|23.047|0.945|0.853|0.078|0.223|0.168|
> |Stone|gsplat+ $L_{\alpha}$|33.832|33.823|0.948|0.948|0.062|0.062|0.891|
> |Stone|NGS (gsplat + $L_{\alpha}$ + noise)|34.148|**34.148**|**0.951**|**0.951**|**0.053**|**0.053**|**0.922**|
> ||
> |OmniObject3D|3DGS|29.3|27.456|0.94|0.929|0.069|0.116|0.215|
> |OmniObject3D|GOF|32.259|24.931|0.97|0.898|0.062|0.122|0.208|
> |OmniObject3D|StopThePop|32.274|25.095|0.97|0.9|**0.05**|0.113|0.265|
> |OmniObject3D|gsplat+ $L_{\alpha}$|33.575|33.350|**0.973**|**0.972**|0.060|0.064|0.642|
> |OmniObject3D|NGS (gsplat + $L_{\alpha}$ + noise)|**33.619**|**33.578**|0.972|**0.972**|0.060|**0.060**|**0.736**|
>
> As the first work to address this problem, **no direct baselines for false transparency exist**. Therefore, we introduced new evaluation tools:
> 1. A **qualitative infill method** to visually highlight transparent surfaces (Fig. 5, S4-6).
> 2. **Standard metrics** (PSNR*, SSIM*, LPIPS*) **computed with infills** to quantify errors in these problematic regions.
> 3. The **quantitative SOS** to measure surface opacity, with higher values indicating less transparency.
>
> Regarding baseline comparisons, Tables 1, S1, and S2 already present comprehensive quantitative evaluations against strong reconstruction methods (original 3DGS, GOF, StopThePop) on three datasets (DTU, OmniObject, Stone), using both standard metrics and our proposed additions.
>
>
> **Q1: How much does false transparency affect rendering quality**
>
> This is explained in Weakness 1.2 above. In short, an opaque object appears as a translucent shell filled with distracting, spiky internal structures that are only visible during interactive camera movement. While we cannot include URLs in the paper, we will include interactive demos on our project page. The effect is readily observable on 3DGS sharing platforms in object-centric models with uniform textures (e.g., food, toys, decorations).
>
> **Q2: qualitative visualizations on standard open datasets.**
>
> We will add more qualitative visualizations for the DTU and OmniObject3D datasets in the revised manuscript, similar to those provided  in Fig. S4-5.
>
> **Q3: whether injected noise infill introduces new artifacts/unintended occlusion, and how it avoids damaging fine surface details.**
>
> This is a critical point. We prevent noise from degrading surface quality through a multi-step process detailed in Sec. 3.3:
> 1. **Depth-aware pruning** (L154): We remove noise Gaussians that are in front of the object's surface based on depth ordering.
> 2. **Binary erosion of the occupancy volume** (L155): We erode the occupancy grid to create a buffer zone, ensuring noise Gaussians do not interfere with surface optimization.
> 3. **Trainable noise Gaussian opacity** (L163): We freeze the surface and train only the opacity of noise Gaussians, which effectively prunes any noise that has become integrated with the surface.
> 4. **Random noise color** (L165): We randomize noise color at each iteration to prevent the model from integrating the noise into the surface texture.
>
> As shown in our ablation study (Table 2), these measures are essential for maintaining rendering quality. We will add a summary at the beginning of Section 3.3 to make this logic clearer.
>
> **Q4: Generalization to more complex scenes and initialization/management of noise infill in such cases.**
>
> We will add a discussion on this in the discussion section. Our method is most relevant for large, well-observed objects, as false transparency is less pronounced in small, thin, or partially-viewed objects. For complex scenes with multiple objects, our approach can be extended. One could use segmentation methods like Segment-Anything [arXiv:2304.02643] to isolate each object, compute its convex hull, and then apply our noise infill and opacity enforcement independently to each significant object in the scene.
>
> **Q5: Typos and figure errors.**
>
> We will proofread the manuscript to correct all typos and figure errors, including the "FTS" vs. "SOS" terminology in our tables. We thank the reviewer and welcome them to point out any other specific errors they may have noticed.

---

> > ### Comment · Reviewer_6z1z · 2025-08-04
> >
> > Thanks for the clarifications. Some terms and descriptions now become clear. At this stage, however, the effectiveness of the proposed method is still not sufficiently supported.

---

> > > ### Author Response · Authors · 2025-08-04
> > > **Clarification on the effectiveness of the proposed method**
> > >
> > > We thank the reviewer for their feedback and the opportunity to clarify our evaluation methodology. We agree that assessing the effectiveness of a method for a previously unreported problem requires careful justification. Here are some points we want to re-emphasize:
> > >
> > >
> > > - **New problem:** A central contribution of our work is the identification and analysis of "false transparency," a new rendering artifact that occurs primarily during interactive viewing when the viewpoint is in motion. There is no previous baseline on false transparency for direct comparison.
> > >
> > > - **Limitation of Standard Metrics:** By its very nature, false transparency is only obvious in interactive viewing, and **cannot be measured** by standard quality metrics (PSNR, SSIM, LPIPS) which are designed for static images. This is why a simple comparison on these metrics alone fails to show the specific problem our work addresses.
> > >
> > > - **New Tools for a new Problem:** To provide an effective assessment of false transparency in static images, we introduced a set of new tools specifically designed to visualize and quantify this artifact:
> > >     1. Our method determines the inner space of an object and creates a noise infill asset
> > >     2. The noise infill can be inserted into a model created by any splatting-based reconstruction method
> > >     3. If the model has a transparent surface, a key symptom of false transparency, the infill will contribute to the final rendering, leading to impaired visuals and lower quality scores against ground truth.
> > >     4. We introduced surface opacity score (SOS) that directly measures the surface opacity, which corresponds to likelihood of false transparency
> > >     5. We prepared a new dataset highlighting the false transparency artifact. We also created noise infill asset for DTU and OmniObject3D to support future benchmark.
> > >
> > > - **Effectiveness of the method:** NGS is explicitly engineered to solve the false transparency problem, rather than to incrementally improve general scores on standard metrics.
> > > Our results demonstrate that our method achieves better results on the infill-augmented metrics (PSNR* , SSIM*, LPIPS*) compared to baseline methods.
> > > Crucially, our results confirm that NGS successfully resolves this targeted issue while maintaining reconstruction quality on par with existing state-of-the-art methods. It is a targeted fix, not a general-purpose enhancement.
> > > Since it is difficult to directly demonstrate this artifact in words and static media, We will also provide interactive visual comparison in the project page that better demonstrates this artifact and our improvement.
> > >
> > > Given this context, our method is design to address the false transparency problem. We would be grateful if the reviewer could suggest any specific analyses or baselines they believe would further strengthen our claims regarding the false transparency artifact.
> > > \
> > > \
> > > \
> > > \
> > > \
> > > **Finally, we would also to highlight that the contribution of the work is not limited to solving the problem:**
> > > 1. We are the first to identify and define "false transparency," a critical rendering artifact in Gaussian Splatting.
> > > 2. We investigated the mechanism of this artifact.
> > > 3. We provide the first set of evaluation tools and datasets to enable future research on this problem.

---

### Official Review · Reviewer_faeg · 2025-07-03

**Clarity:** 3
**Significance:** 3
**Originality:** 3
**Rating:** 5
**Confidence:** 3

**Summary:**

This paper provides a plug and play solution to mistakes in transparency estimation of gaussian splatting in an object oriented setup.
Given a segmentation mask of the object surface, the method proposes to add noisy opaque gaussians inside the object to break the line of sight between different viewpoints and force the optimization to focus on the surface.
For the initialization they run a convex hull mesh algorithm on the initial gaussians and the prune it based on the depth ordering of the gaussians.
They provide a method for better evaluation of the transparency issue in 3dgs by doing green infill and show the noise guided splatting improves significantly upon the baselines and similar methods.

**Questions:**

Why is the pruning kept in the method?

**Ethical Concerns:**

["NO or VERY MINOR ethics concerns only"]

**Final Justification:**

I have read the other reviewer's concerns regarding the impact or importance of these false transparencies. In my experience they are quite important if you are to use 3dgs as a true 3d model rather than just a video of camera trajectory generator, such as for VR and Gaming.

It is unfortunate that currently we don't have evaluation metrics that is more comprehensive regarding the full 3D model of a scene, which should be a point of further investigation and benchmarking in future. Ideally we should be able to quantify the effect of pruning on the results easily, but I agree that is not the scope of this project.

Hence, I keep my rating as accept.

**Limitations:**

yes

**Quality:**

3

**Strengths And Weaknesses:**

Strengths
The method is novel with a small amount of overhead and easily applicable to different gaussian splatting methods.
The experiments are conclusive over 3 datasets and ablations cover the different elements of the method.
The paper is well written.

Weaknesses:
Based on the ablation it seems that pruning is not helpful so the justification for keeping it is unclear.

---

> ### Author Rebuttal · Authors · 2025-07-31
>
> We sincerely thank you for your positive review and for highlighting the novelty, applicability, and conclusive experiments of our method. We are particularly grateful for your appreciation of our quantitative evaluation approach for false transparency.
>
> **Question: Why is the pruning kept in the method?**
>
> This is an excellent question. We acknowledge that the impact of removing pruning is not obvious from the quantitative metrics in our ablation study (Table 2), a point we will clarify in the manuscript.
>
> While the global metrics remain high, pruning is crucial for visual quality, particularly in regions with high geometric complexity or concavity. Without pruning (specifically, the depth-aware pruning and erosion steps in Sec. 3.3), we observed that noise Gaussians can become embedded within the surface layer itself. This creates small but visually distracting artifacts like blurring or color blending on the object's surface.
>
> These localized artifacts are not well captured by global metrics like PSNR or SSIM, which average errors across the entire image. However, they cause small local blurs of the reconstructed object.
>
> Therefore, we retain pruning not for a major quantitative boost, but to guarantee the visual fidelity and crispness of the surface. It ensures that noise is strictly confined to the object's interior and does not interfere with the foreground, which is a key goal of our work.
>
> To make this clearer, we will update the revised manuscript to:
> - Clarify this reasoning in the ablation study (Sec. 4.4).
> - Include a visual comparison showing these subtle artifacts.
> - Briefly justify the need for pruning in the method description (Sec. 3.3).
>
> We hope this clarifies the importance of the pruning step. Thank you once again for your constructive feedback.

---

### Note · Authors · 2025-08-12

Our work reveals a new false transparency artifact in 3DGS interactive viewing, undetectable in static renderings with standard metrics. We provide a theory for its root cause and an effective cure. We also provide new tools to detect/quantify this artifact in static renderings, enabling assessment during training/evaluation. We didn’t clearly define the symptoms of the artifact and misspelled a critical new metric SOS as FTS, which caused major confusion. Our revision addressed these issues.
\
\
**faeg** recognized our contributions and only had a minor issue on a pruning implementation, which is clarified.
\
\
**6z1z** questioned the method’s effectiveness. All other reviewers explicitly found it effective:
- faeg: “improves significantly upon the baselines and similar methods”
- N48H: “the method is proven to be effective”
- BBSm: “An effective algorithm to address it”

The difference could arise from evaluation criteria. False transparency does not affect traditional metrics (SSIM/PSNR) or static images. Its symptom is only visible during interactive viewing, where dynamic parallax reveals transparent surfaces. A core contribution is providing new tools (SOS, metrics w/ infill, datasets) to detect/quantify the artifact in static views and training. NGS effectively improved new metrics without affecting standard metrics.
\
\
**N48H** noted we effectively addressed false transparency and provided new tools (metrics and datasets) to support diagnostics, but felt we missed theoretical basis, relied on assumptions, and showed limited broader NVS gains.

NGS is based on a strong theoretical analysis of the artifact’s root cause: the ill-posed alpha blending. The theory explains why such “assumptions” are conditions for false transparency. NGS does not apply to some scenarios (e.g., thin structures, partial views) simply because no background contributes to the blending.
Arguing that impact requires NVS gains misses the point. The artifact regularly occurs in object centric scans, yet standard metrics fail to capture it - prompting us to develop new tools.

Object scans are central to 3D assets in games, films, ads, and VFX. Large datasets like uCO3D and Project Aria are devoted to it. As demand for 3DGS assets rapidly grows, solving false transparency is essential for improving interactive visual quality and facilitating adoption.
\
\
**BBSm** confused the artifact with floaters, but recognized the meaningful technical contributions after our clarification.

---

### Decision · Program_Chairs · 2025-09-17

**Decision:**

Accept (poster)

**Comment:**

This paper presents a simple, yet highly effective and practical solution to a specific but important problem in 3D Gaussian Splatting (3DGS): erroneous transparency in object-centric scenes. The core idea—strategically seeding opaque, noisy Gaussians within an object's volume to break the line-of-sight between viewpoints—is both novel and intuitive. This approach cleverly forces the optimization to converge on a correct, opaque surface representation.

A significant strength of this work is its practical utility. The method is designed as a plug-and-play module with low computational overhead, making it easily integrable into existing 3DGS pipelines. This usability enhances its potential impact on the community.

The experimental validation is thorough and convincing. Results across three datasets demonstrate a clear and significant improvement over baseline methods. The introduction of a green infill evaluation metric is also a valuable contribution, providing a clear and direct visual method for assessing transparency artifacts. The ablation studies are generally well-designed, effectively isolating the contribution of the core "noisy splatting" mechanism.

The noted weakness—that the pruning step appears unhelpful in the ablation—is a valid observation. However, this is a minor flaw in an otherwise strong paper. The retention of this step does not detract from the primary contribution, which is the overall initialization and optimization strategy. The core concept remains sound and impactful without it.

In conclusion, this work addresses a well-defined problem with an elegant, effective, and highly applicable solution. The novelty, practical utility, and strong empirical results make it a valuable contribution worthy of acceptance. The minor issue with the pruning step can be addressed by the authors in a revision, perhaps by providing a clearer justification or removing the step altogether.